# linc-mipep and linc-wrb encode micropeptides that regulate chromatin accessibility in vertebrate-specific neural cells

Valerie A Tornini[1]*, Liyun Miao[1†], Ho-Joon Lee[1,2†], Timothy Gerson[1‡], Sarah E Dube[1‡], Valeria Schmidt[1], François Kroll[3], Yin Tang[1], Katherine Du[1,4], Manik Kuchroo[1,4], Charles E Vejnar[1], Ariel Alejandro Bazzini[5,6], Smita Krishnaswamy[1,4], Jason Rihel[3]*, Antonio J Giraldez[1,7,8]*

[1]Department of Genetics, Yale University, New Haven, United States; [2]Yale Center for Genome Analysis, Yale University, New Haven, United States; [3]Department of Cell and Developmental Biology, University College London, London, United Kingdom; [4]Department of Computer Science, Yale University, New Haven, United States; [5]Stowers Institute for Medical Research, Kansas City, United States; [6]Department of Molecular & Integrative Physiology, University of Kansas School of Medicine, Kansas City, United States; [7]Yale Stem Cell Center, Yale University School of Medicine, New Haven, United States; [8]Yale Cancer Center, Yale University School of Medicine, New Haven, United States

*For correspondence:
valerie.tornini@yale.edu (VAT);
j.rihel@ucl.ac.uk (JR);
antonio.giraldez@yale.edu (AJG)

[†]These authors contributed equally to this work
[‡]These authors also contributed equally to this work

**Abstract** Thousands of long intergenic non-coding RNAs (lincRNAs) are transcribed throughout the vertebrate genome. A subset of lincRNAs enriched in developing brains have recently been found to contain cryptic open-reading frames and are speculated to encode micropeptides. However, systematic identification and functional assessment of these transcripts have been hindered by technical challenges caused by their small size. Here, we show that two putative lincRNAs (linc-mipep, also called lnc-rps25, and linc-wrb) encode micropeptides with homology to the vertebrate-specific chromatin architectural protein, Hmgn1, and demonstrate that they are required for development of vertebrate-specific brain cell types. Specifically, we show that NMDA receptor-mediated pathways are dysregulated in zebrafish lacking these micropeptides and that their loss preferentially alters the gene regulatory networks that establish cerebellar cells and oligodendrocytes – evolutionarily newer cell types that develop postnatally in humans. These findings reveal a key missing link in the evolution of vertebrate brain cell development and illustrate a genetic basis for how some neural cell types are more susceptible to chromatin disruptions, with implications for neurodevelopmental disorders and disease.

## Editor's evaluation

The study describes the discovery of two related micro-peptides that regulate zebrafish behavior by affecting chromatin accessibility in the embryonic brain. Zebrafish mutants lacking these micropeptides show altered gene regulatory networks that preferentially affect oligodendrocytes and cerebellar cells in the embryonic brain. The data presented in the study is solid and presents convincing additional evidence for versatile functions of micro-peptides.

## Introduction

While most of the vertebrate genome is transcribed, only a small portion encodes for functional proteins. Much of the remaining transcriptome is comprised of non-coding RNAs, including thousands of predicted long intergenic non-coding RNAs (lincRNAs). Despite this large number of lincRNAs, the functional significance of most remains unclear (*Goudarzi et al., 2019*). Recent advances in ribosome profiling and mass spectrometry have identified short open-reading frames (sORFs) within putative lincRNA sequences that may encode micropeptides, which were otherwise missed due to their small size (<100 aa) (*Bazzini et al., 2014*; *Chen et al., 2020*; *Ingolia et al., 2009*; *Kondo et al., 2007*; *Pauli et al., 2014*; *Couso and Patraquim, 2017*). Despite conventional rules assuming that short peptides are unlikely to fold into stable structures to perform functions and subjective cut-offs (100 aa) used in computational identification of protein coding genes, there are several examples of these small peptides performing diverse, important cellular functions (*Bi et al., 2017*; *Chen et al., 2020*; *D'Lima et al., 2017*; *Fields et al., 2015*).

Many lincRNAs are expressed in a tissue-specific manner, and about 40% of all long noncoding RNAs identified in the human genome are specifically expressed in the central nervous system (*Derrien et al., 2012*; *Ulitsky et al., 2011*). The vertebrate central nervous system consists of some of the most diverse and specialized cell types in the vertebrate body and has distinct chromatin states and gene regulatory networks that have evolved to establish and maintain this diversity. Since many micropeptides have a relatively recent evolutionarily origin and, given their small size, may be able to access and regulate cellular machines inaccessible by larger proteins (*Makarewich and Olson, 2017*), the lincRNA tissue-specificity may indicate undiscovered roles in vertebrate-specific CNS development and function.

Evolutionarily recent micropeptides may contribute to vertebrate-specific functions and phenotypes that have otherwise been missed due to misclassification as non-coding transcripts and lack of high-throughput phenotyping for coding functions. We sought to identify micropeptides that were cryptically encoded in long non-coding RNAs but were missed due to assumptions about minimal protein sizes, dubious homologies, or mis-annotations. Here, we interrogate the function of predicted non-coding RNAs and identify two related micropeptides that regulate behavior, chromatin accessibility, and gene regulatory networks that establish evolutionarily newer neural cell types.

## Results

### Screen of long non-coding RNAs identifies micropeptide regulators of vertebrate behavior

To identify lincRNAs that may encode for micropeptides, we first analyzed ribosome profiles for previously published lincRNAs *Ulitsky et al., 2011* in zebrafish embryos during early development (0–48 hr post-fertilization) (*Bazzini et al., 2014*), performed in situ hybridization on 21 of these candidates, and identified brain-enriched micropeptide candidates (*Figure 1—figure supplement 1*; *Supplementary file 1*). To identify the physiological role of ten of these putative micropeptides, we adapted an F0 CRISPR/Cas9 behavioral screening pipeline (*Figure 1A*; *Kroll et al., 2021*). CRISPR/Cas9 targeting efficiently induced a range of mutations in the targeted gene sequences, with inferred indel or large deletion rates with multiple guides estimated between ~40 and 100% per targeted locus, including frame-shift mutations (*Figure 1—figure supplement 2*; *Supplementary file 1*).

At 4–7 days post-fertilization (dpf), zebrafish display a repertoire of conserved, stereotyped baseline locomotor behaviors across day:night cycles (*Prober et al., 2006*; *Rihel et al., 2010*; *Kroll et al., 2021*). To quantitatively track locomotor activity of wild type and F0 mutant fish, single larvae from each condition were placed into individual clear wells of a clear 96-square well flat plate, then placed on a tracking platform that detects the change in pixels per frame for each well, between 4 dpf and 7 dpf (*Figure 1A*). We measured daytime and nighttime behavioral parameters, calculated the deviation (Z-score) of each F0 mutant larva from its wild type siblings, generated 'behavioral fingerprints' (*Figure 1—figure supplement 3A*), and measured the Euclidean distance between each larva and the mean fingerprint of its wild type siblings (*Figure 1—figure supplement 3B*).

This screen identified two candidate genes, *linc-mipep* and *linc-wrb,* that had a specific daytime hyperactivity phenotype and correlated behavioral fingerprints (*r*=0.67) when mutated in the ORF identified by ribosome footprints (*Figure 1B*; *Figure 1—figure supplement 3C*; *Figure 1—figure*

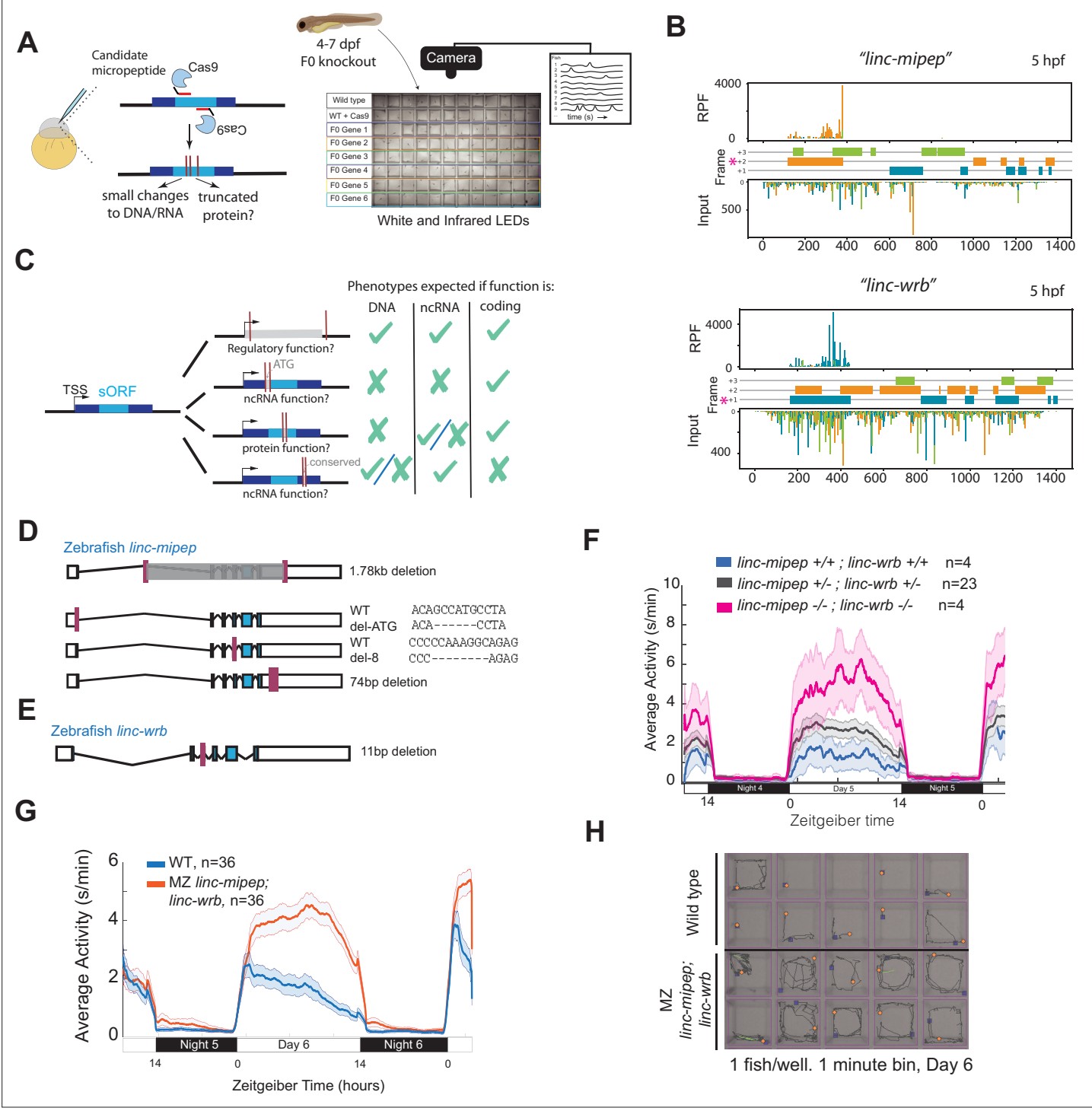

**Figure 1.** *linc-mipep* and *linc-wrb* loss-of-protein-function mutant larvae are behaviorally hyperactive. (**A**) Left, schematic of F0 CRISPR knockout behavioral screen. Zebrafish embryos were injected early at the one-cell embryo stage with multiple sgRNAs and Cas9 targeting the ORF of candidate micropeptides encoded within putative lincRNAs. Right, schematic of behavior screening platform. Each well of a 96-well flat bottom plate contains one zebrafish larva (4–7days post-fertilization, dpf) from the same wild type (WT) clutch. Individual locomotor activity was tracked at 25 frames per second on a 14hr:10hr light:dark cycle. (**B**) Ribosome footprint of *linc-mipep* (also known as *lnc-rps25*) (top) or *linc-wrb* (bottom) at 5hours post fertilization (hpf) across annotated transcript length, with putative coding frames in green (+3), orange (+2), or blue (+1); input (control) on bottom tracks. Magenta asterisk marks predicted short open reading frame. RPF, ribosome-protected fragment. (**C**) Summary of mutagenesis strategy to decode transcript functions. Magenta bars denote CRISPR-targeted area. Mutated/removed sequence is in gray. TSS, transcription start site. sORF, short open reading frame. ATG, start codon. ncRNA, non-coding RNA. Right, phenotypes predicted (check mark) or not predicted (x mark) for each mutant if the gene

*Figure 1 continued on next page*

*Figure 1 continued*

functions as a regulatory region, noncoding RNA, or protein-coding gene. (**D**) Stable mutants for *linc-mipep:* full region deletion (1.78kb deletion, from intron 1 – proximal 3'UTR, top); translation start site deletion that removes the ATG sequence (middle); frameshift deletion (8bp deletion at exon 4, second from bottom); 74bp deletion that removes highly conserved 3'UTR sequence (bottom). (**E**) Stable frameshift mutant for *linc-wrb* (11bp deletion, exon 3). (**F**) Locomotor activity of *linc-mipep*$^{del-1.8kb/del-1.8kb}$;*linc-wrb*$^{del-11/del-11}$ (*linc-mipep -/-; linc-wrb -/-*, magenta); *linc-mipep*$^{del-1.8kb/+}$;*linc-wrb*$^{del-11/+}$ (*linc-mipep +/-; linc-wrb +/-*, black); and wild-type (*linc-mipep +/+; linc-wrb +/+*, blue) sibling-matched larvae over 2 nights. (**G**) Locomotor activity of wild type (WT, blue) or maternal-zygotic *linc-mipep*$^{del1.8kb/del1.8kb}$;*linc-wrb*$^{del11bp/del11bp}$ (*linc-mipep;linc-wrb,* orange) larvae across two nights. The ribbon represents± SEM. Zeitgeber time is defined from lights ON = 0. (**H**) Representative daytime locomotor activity tracking of wild type (top 2 rows) and maternal-zygotic *linc-mipep*$^{del1.8kb/del1.8kb}$;*linc-wrb*$^{del11bp/del11bp}$ (*linc-mipep;linc-wrb,* bottom 2 rows) larvae during 1min at 6 dpf. Blue and orange dots represent start and stop locations, respectively.

The online version of this article includes the following figure supplement(s) for figure 1:

**Figure supplement 1.** Expression of micropeptide candidates.

**Figure supplement 2.** Validation of CRISPR targeting in F0 screen.

**Figure supplement 3.** Screening for micropeptide loss-of-function effects on zebrafish baseline behavior.

**Figure supplement 4.** Average daytime activity differences between WT and F0 knockouts of candidate micropeptides.

**Figure supplement 5.** *linc-mipep* and *linc-wrb* gene expression in early zebrafish development.

**Figure supplement 6.** Stable *linc-mipep* and *linc-wrb* mutant behavioral profiles and sequence verification.

**Figure supplement 7.** *linc-mipep* and *linc-wrb* loss-of-protein-function mutant larvae are behaviorally hyperactive in a dose-dependent manner.

*supplement 4*). Sequence analysis revealed that *linc-mipep* (current nomenclature *si:ch73-1a9.3*, ENSDART00000158245, also called *lnc-rps25*) and *linc-wrb* (current nomenclature *si:ch73-281n10.2*, ENSDART00000155252) (***Bazzini et al., 2014***; ***Ulitsky et al., 2011***; ***Figure 1—figure supplement 5A, B***) both had homology in their sORFs' exon structure (***Figure 1—figure supplement 5A, B***) and mRNA sequences (BLAST identity score = 72%) (***Supplementary file 1***), as well as a highly conserved element (92% identical sequence) in their non-coding sequences (***Figure 1—figure supplement 5C***). While both *linc-mipep* and *linc-wrb* were originally identified as long non-coding RNAs, both genes have ribosome-protected fragments, suggesting they are likely encoding proteins 87aa and 93aa in size, respectively (***Figure 1B***; ***Figure 1—figure supplement 5D, E***). In situ hybridization and RNA-sequencing revealed that transcripts for both genes are expressed throughout embryogenesis, through 5 dpf (***Figure 1—figure supplement 5F, G***). These results indicate that *linc-mipep and linc-wrb* might encode redundant or paralogous genes functioning as either lincRNAs or micropeptide-encoding genes involved in behavior.

## *linc-mipep* and *linc-wrb* encode for related micropeptides that regulate zebrafish behavior

Although *linc-mipep* and *linc-wrb* are transcribed and likely translated (***Figure 1B***; ***Figure 1—figure supplement 5D–F***), ribosome profiling data is insufficient to distinguish between pervasive background translation and translation of functional proteins. For example, these could represent sORFs within enhancer RNAs or in noncoding RNAs that have acquired an ORF but yield a nonfunctional protein. Thus, to distinguish whether *linc-mipep* and *linc-wrb* function as regulatory DNA, noncoding RNA, or protein coding genes, we used CRISPR-Cas9 gene editing to generate stable deletion mutants that either target the full sequence, the translation start site, the putative coding region, or the conserved untranslated/non-coding region (***Figure 1C–E*** ; ***Figure 1—figure supplement 6***). Examining the behavioral profile of these mutants identified a consistent and specific increase in locomotor activity during the daytime in all mutants affecting the ORF for both *linc-mipep* and *linc-wrb* (***Figure 1—figure supplement 6A–D***). In contrast, deleting the highly conserved element in the untranslated region in *linc-mipep*, which could encode a conserved lincRNA sequence, did not result in any detectable morphological or behavioral phenotypes (***Figure 1—figure supplement 6E***).

First, we asked whether the coding part of these genes is necessary. Start codon mutations in *linc-mipep* (zygotic or maternal-zygotic *linc-mipep*$^{ATG-del6}$) resulted in a similar daytime hyperactivity phenotype as frameshift mutations (*linc-mipep*$^{del8}$) or deletion of most of the *linc-mipep* region (*linc-mipep*$^{del1.8kb}$) (***Figure 1—figure supplement 6A–C***, ***Figure 1—figure supplement 7A***). These results indicate that the observed phenotypes are the result of protein coding function of *linc-mipep* rather than a non-coding transcript or a regulatory DNA sequence function. Double *linc-mipep*$^{del1.8kb}$;

*linc-wrb$^{del11}$* homozygous mutants display even higher daytime locomotor hyperactivity levels compared to *linc-mipep; linc-wrb* heterozygous or wildtype larvae (*Figure 1—figure supplement 7C*), with no significant changes in nighttime activity (*Figure 1F*), a phenotype that is maintained if we remove the maternal contribution in maternal-zygotic (MZ) *linc-mipep$^{del1.8kb}$; linc-wrb$^{del11}$* animals (*Figure 1G and H*; *Figure 1—figure supplement 7B*). Each additional loss of a copy of either gene generally results in higher hyperactivity levels (*Figure 1—figure supplement 7C*), suggesting that these genes may work together in a dose-dependent manner.

Next, we asked whether the coding part of these genes is sufficient to drive behavior. To determine that the behavioral phenotypes observed in mutants result from the loss of coding function, we generated transgenic zebrafish that ubiquitously express the coding sequence (CDS) of *linc-mipep* and tracked their behavior (*Figure 2A and B*). The sORF encoded in *linc-mipep* was able to rescue the hyperactivity phenotypes in *linc-mipep* mutants (*Figure 2C and D*) without significant changes to wild type activity levels (*Figure 2D*) or in nighttime activity (magnified, *Figure 2C*). Moreover, *linc-mipep* expression was able to rescue the hyperactivity of *linc-wrb* heterozygous mutants to almost wild type levels (*Figure 2—figure supplement 1A, B*), suggesting that these proteins share properties that can rescue loss of the other.

Finally, to confirm and visualize the protein encoded by *linc-mipep* and *linc-wrb*, we developed custom antibodies (*Figure 2—figure supplement 2A, B*). The protein product of both transcripts are detected in developing wild type embryos and larvae (*Figure 2—figure supplement 2C–K*). We find that these proteins are expressed throughout early development, with stronger staining and broader expression pattern for the protein encoded by *linc-mipep* compared to that of *linc-wrb* (*Figure 2—figure supplement 2G, H*). We note nonspecific staining of the Linc-wrb antibody in embryos and in likely endothelial cells throughout early development, as staining is still detected in these cells in *linc-wrb* mutants (*Figure 2—figure supplement 2I, K*). We further observed that the protein products of both transcripts are enriched in non-dividing wild-type nuclei (*Figure 2—figure supplement 2J, K*) and absent in *linc-mipep;linc-wrb* loss-of-function mutant embryos (*Figure 2—figure supplement 2J, K*) and larval brains (*Figure 2—figure supplement 2L, M*). Together, these results indicate that *linc-mipep* and *linc-wrb* encode for nuclear-localized micropeptides that have a dosage effect to regulate locomotor activity and behavior in zebrafish.

## Vertebrate-specific evolutionary and functional conservation of proteins encoded by *linc-mipep* and *linc-wrb*

Protein BLAST of both Linc-mipep (87aa) and Linc-wrb (93aa) ORFs identified conserved sequences across teleosts and other vertebrates, including humans, with homology to non-histone chromosomal protein HMG-14, or High Mobility Group N1 (HMGN1), and the related HMG-17/HMGN2 protein (*Supplementary file 2*; *Bustin, 2001*). Whereas the cDNA sequence showed some mild conservation (*Figure 2—figure supplement 3A*), the highly conserved proximal 3'UTR elements instead allowed us to identify homologous predicted lincRNAs, unannotated genes, pseudogenes, and HMGN1 genes across vertebrate species spanning over 450 million years (*Figure 2—figure supplement 3B–C*; *Figure 2—figure supplement 4*; *Supplementary file 2*; *Kumar and Hedges, 1998*).

We first identify that *linc-wrb* is syntenic to human *Hmgn1* (*Figure 2—figure supplement 5D–F*). To identify the evolutionary origin of this gene, we traced back the synteny for sequences or expressed sequence tags (ESTs) that were identified between flanking genes that are syntenically conserved with humans, *Get1/Wrb* and *Brwd1* (*Figure 2—figure supplement 5E*). Through these analyses, we were first able to identify an unannotated ORF in the basal agnathan (jawless vertebrate) lamprey, syntenic to human HMGN1, that encodes for an ancestral protein more similar to human HMGN2 (*Figure 2—figure supplement 5B*; *Figure 2—figure supplement 3E*). Though we did not identify any *linc-mipep* or *linc-wrb* protein-coding homolog in invertebrates (*Supplementary file 2*), in line with previous results (*Johns, 1982*), we did identify an ORF syntenic to *linc-wrb* in the invertebrate basal chordate lancelet (or amphioxus) genome (*Figure 2—figure supplement 5A*). When we analyzed whether there were any similarities between the sequence of this APEX1-like gene in the lancelet genome (*Figure 2—figure supplement 5A*), we found by BLAST that its N-terminal sequence (30aa) aligns to HMGN family members in various vertebrate species (*Figure 2—figure supplement 3D*). These results suggest that the N-terminal sequence of the gene in the ancestral location that would

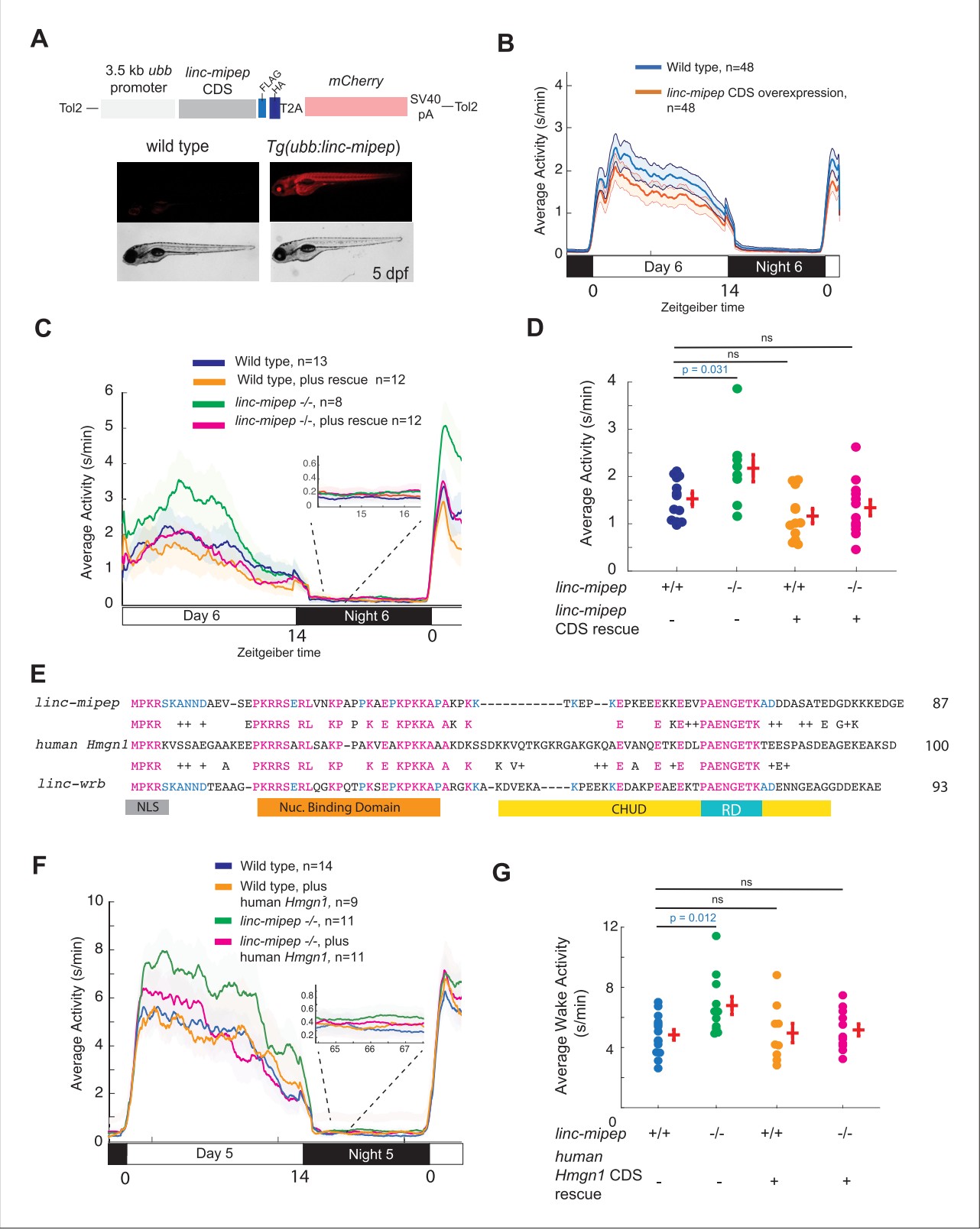

**Figure 2.** *linc-mipep* and *linc-wrb* encode proteins with homology to human HMGN1. (**A**) Top, diagram of transgenic *linc-mipep* overexpression construct. Transgenic lines were established via Tol2-mediated integration of 3.5kb *ubiquitin B (ubb)* promotor driving the *linc-mipep* coding sequence with a FLAG and HA tag at the C-terminus, followed by a T2A self-cleaving peptide, *mCherry* reporter, and SV40 polyA tail. Bottom, fluorescent and brightfield images of 5 dpf zebrafish siblings either without overexpression (wild type, *mCherry*-negative, left) or with *linc-mipep* overexpression

*Figure 2 continued*

(*mCherry*-positive, right). (**B**) Activity plot of wild type (mCherry-negative, blue) or *linc-mipep* overexpression (*Tg(ubb:linc-mipep)* mCherry-positive, orange) siblings at 6 dpf. n=48 per genotype. Average day activity p=0.028, one-way ANOVA. (**C**) Locomotor activity of *linc-mipep* mutants, with or without transgenic *linc-mipep* overexpression (*Tg(ubb:linc-mipep CDS-T2A-mCherry)*, 'rescue'), sibling-matched larvae over 24hr. Inset, no effect on nighttime activity. (**D**) Average waking activity of 6 dpf *linc-mipep* mutant, heterozygous, or wild type larvae, with (denoted by +) or without (denoted by -) *linc-mipep* transgenic rescue. Each dot represents a single fish, and crossbars plot the mean ± SEM. p Values from a Dunnett's test, using wild type (*linc-mipep +/+*) as the baseline condition. (**E**) Amino acid sequences of *linc-mipep* (top), *linc-wrb* (bottom), and human *Hmgn1* (middle). Conserved amino acids are denoted in blue (if conserved between two sequences) or magenta (if conserved across the three sequences). Conserved functional domains for Hmgn1 are denoted (NLS, nuclear localization signal; Nuclear Binding Domain; RD, Regulatory Domain; and CHUD, Chromatin Unwinding Domain). (**F**) Locomotor activity of *linc-mipep* mutants, with or without transgenic human *Hmgn1* overexpression (*Tg(ubb:hHmgn1CDS-T2A-mCherry)*, 'rescue'), sibling-matched larvae over 24hr. Inset, no effect on nighttime activity. (**G**) Average waking activity of 6 dpf *linc-mipep* mutant (-/-) or wild type (+/+) larvae, with (denoted by +) or without (denoted by -) human *Hmgn1* transgenic rescue. Each dot represents a single larva, and crossbars plot the mean ± SEM. p Values from a Dunnett's test, using wild type (*linc-mipep +/+*) as the baseline condition.

The online version of this article includes the following figure supplement(s) for figure 2:

**Figure supplement 1.** *linc-wrb* mutant behavioral phenotype can be rescued by transgenic *linc-mipep* CDS, though not by human Hmgn1 CDS.

**Figure supplement 2.** Antibody staining confirmation of proteins encoded by *linc-mipep* and *linc-wrb*.

**Figure supplement 3.** *linc-mipep* and *linc-wrb* encode proteins with homology to human HMGN1.

**Figure supplement 4.** Amino acid sequence alignment for identified HMGN1 sequences across species.

**Figure supplement 5.** Syntenic analysis of *linc-mipep* and *linc-wrb*.

**Figure supplement 6.** Relationships between genes encoded by *linc-mipep* and *linc-wrb* across fish species.

give rise to *linc-wrb* and human HMGN1 may have been co-opted to give rise to the HMGN gene and pseudogene families in vertebrates.

We next searched for the evolutionary origins of *linc-mipep*. The highly conserved 3'UTR suggested that *linc-mipep* and *linc-wrb* derived from the same ancestral gene, either before or after the teleost-specific genome duplication. To address this question, we analyzed the regions syntenic to human HMGN1 in spotted gar, a slowly evolving species whose lineage diverged from teleosts before the teleost genome duplication, and in coelacanth, a lobe-finned fish with the slowest evolving bony vertebrate genome that split from ray-finned fish such as gar and zebrafish (*Braasch et al., 2016*). In coelacanth, we only identified one protein sequence that aligns to HMGN1, syntenic to human HMGN1, with no ESTs or other sequences identified elsewhere (*Supplementary file 2*). In spotted gar, we found the gene syntenic to *linc-wrb* and human *Hmgn1* (*Figure 2—figure supplement 5C*). Although we did not identify a gene syntenic to *linc-mipep*, we did identify the appearance of both *Mipep* next to *Brwd1*, and of *Igsf5* next to *Sh3bgr* (*Figure 2—figure supplement 5C*). When analyzed compared to the genomic location of *linc-mipep* in zebrafish (*Figure 2—figure supplement 5D*), we suggest that *linc-mipep* may have resulted from a gene duplication of *linc-wrb* into the neighboring IGSF5 intronic region, which then rearranged to land next to *Mipep* in the teleost genome duplication (compare *Figure 2—figure supplement 5C, D*). We found that *linc-mipep* has been maintained in other teleost fish species (*Figure 2—figure supplement 5*; *Figure 2—figure supplement 6*). Together, these findings suggest that *linc-mipep* arose from a gene duplication from *linc-wrb*, and that *linc-wrb* arose from what we identify here as the basal HMGN gene in agnathan lineages.

Finally, to understand whether the proteins encoded by *linc-mipep* and *linc-wrb* share common functions with human Hmgn1, we asked whether the human HMGN1 homologous protein can rescue the hyperactivity of *linc-mipep* and *linc-wrb* mutants. We generated transgenic zebrafish that ubiquitously express the coding sequence (CDS) of human HMGN1 in each mutant background. Human HMGN1 was able to rescue the hyperactivity phenotypes in *linc-mipep* mutants (*Figure 2F and G*), without significant changes in nighttime activity (magnified, *Figure 2F*). We were unable to rescue the *linc-wrb* mutant phenotype with human HMGN1 (*Figure 2—figure supplement 1D*). These data suggest that genes encoded within *linc-mipep* and *linc-wrb* have some functional homology with each other, and that at least the protein encoded by *linc-mipep* has functional homology with, and can be rescued by, human HMGN1. Based on these results, we propose renaming *linc-wrb* as *hmgn1a*, and *linc-mipep* as *hmgn1b*, as their official nomenclature.

## *linc-mipep; linc-wrb* mutants have dysregulation of NMDA receptor-mediated signaling and immediate early gene induction

To gain insight into pathways regulated by *linc-mipep* and *linc-wrb*, we analyzed the behavioral fingerprints of each mutant compared to zebrafish larvae treated with 550 psychoactive drugs that affect different pathways (*Rihel et al., 2010*). We used hierarchical clustering (*Rihel et al., 2010*) to identify drugs that elicit a similar behavior to the *linc-mipep* and *linc-wrb* mutants (i.e. drugs that phenocopy across multiple day-night behavioral measurements) (*Figure 3—figure supplement 1A, B*, overlapping hits in blue text). We found that *linc-mipep* mutant behaviors most resembled those of WT fish treated with an NMDA receptor antagonist (*Figure 3A*), suggesting that NMDA signaling may be reduced in *linc-mipep* mutants. The *linc-mipep* and *linc-wrb* mutant phenotypes also resembled that of WT fish treated with glucocorticoid receptor activators (*Figure 3A*, *Supplementary file 3*), suggesting that downstream glucocorticoid signaling may be upregulated in the mutants.

The identified drugs may alter either common or parallel pathways as loss of *linc-mipep*. To distinguish between these possibilities, we first assessed the effect of glucocorticoid receptor agonist flumethasone on *linc-mipep* mutant behavior. These treatments further exacerbated the daytime locomotor activity of *linc-mipep*$^{-/-}$ larvae above the control-treated *linc-mipep* mutant levels (*Figure 3—figure supplement 2A*), with higher nighttime activity levels in *linc-mipep* mutants treated with flumethasone (*Figure 3—figure supplement 2B*). Since both the daytime and nighttime effects of glucocorticoids were much stronger in the mutants than in similarly treated wild type controls, *linc-mipep* mutants are sensitized to glucocorticoid signaling. We found similar glucocorticoid sensitivity in *linc-wrb* mutants (*Figure 3—figure supplement 2C, D*).

Next, to test the NMDA receptor pathway, we compared the response of WT and *linc-mipep* mutant to L-701–324, an NMDA receptor antagonist at the glycine binding site. L-701–324 elicited a daytime locomotor hyperactivity in WT larvae to a level that was similar to that of *linc-mipep* mutant larvae and *linc-mipep* larvae treated with L-701–324 (*Figure 3B*). Yet, treatment with higher doses of L-701–324 did not affect or exacerbate the activity levels in *linc-mipep* mutants (*Figure 3C*). We found similar results with *linc-wrb* mutants treated with L-701–324 (*Figure 3—figure supplement 2E, F*). These non-additive results indicate that NMDA receptor antagonism and mutations in *linc-mipep* and *linc-wrb* share a common mechanism for inducing hyperactivity.

## *linc-mipep* and *linc-wrb* regulate chromatin accessibility for transcription factors modifying neural activation

Given that linc-mipep and *linc-wrb* have protein domains with homology to nucleosome binding and chromatin unwinding domains of HMGN1 (*Cuddapah et al., 2011*; *Deng et al., 2013*), and given that both NMDA antagonism and glucocorticoid signaling alter immediate early gene expression, we hypothesized that the daytime hyperactivity might be due to altered chromatin accessibility in the mutants. To test the effect of full loss-of-function of both related proteins encoded by *linc-mipep* and *linc-wrb* on chromatin accessibility, we performed omni-ATAC-seq (*Corces et al., 2017*) at 5 dpf comparing WT and double mutant brains (*Figure 3—figure supplement 3A, B*).

We first observed a broad dysregulation of chromatin accessibility, with 2167 regions losing accessibility and 1220 regions gaining accessibility in *linc-mipep;linc-wrb* mutant brains (*Figure 3D*; *Supplementary file 4*), with most regions remaining unchanged (*Figure 3—figure supplement 3C*). CTCF/L transcription factor (TF) motifs were enriched in regions that lost accessibility, suggesting a possible dysregulation of 3D chromatin structure (*Figure 3E*). Enriched TF motifs at regions that lost accessibility were members of the ATF (activating transcription factor)/CREB (cAMP responsive element binding proteins) family, and AP-1 transcription factor components (*Figure 3E*, left panel). TFs binding at these motifs regulate the expression of immediate early response genes (IEG) such as *c-fos*, *c-jun*, and *c-myc* (*Sheng and Greenberg, 1990*). We confirmed reduced *c-fos* transcription in *linc-mipep;linc-wrb* brains at this timepoint by in situ hybridization and by qPCR (*Figure 3—figure supplement 3D*). We also found that the motifs for the glucocorticoid modulatory element binding protein 2 (GMEB2), and for interferon-stimulated transcription factor 3, gamma (ISGF3G, also called IRF-9), were enriched in regions that lost accessibility in *linc-mipep; linc-wrb* mutants. On the other hand, TFs most enriched in regions that gained accessibility were KLF/SP family members, which promote stem cell pluripotency and are downregulated during differentiation, and EGR family members (*Figure 3E*, right panel; *Yamane et al., 2018*). Altogether, these results indicate that *linc-mipep; linc-wrb* have

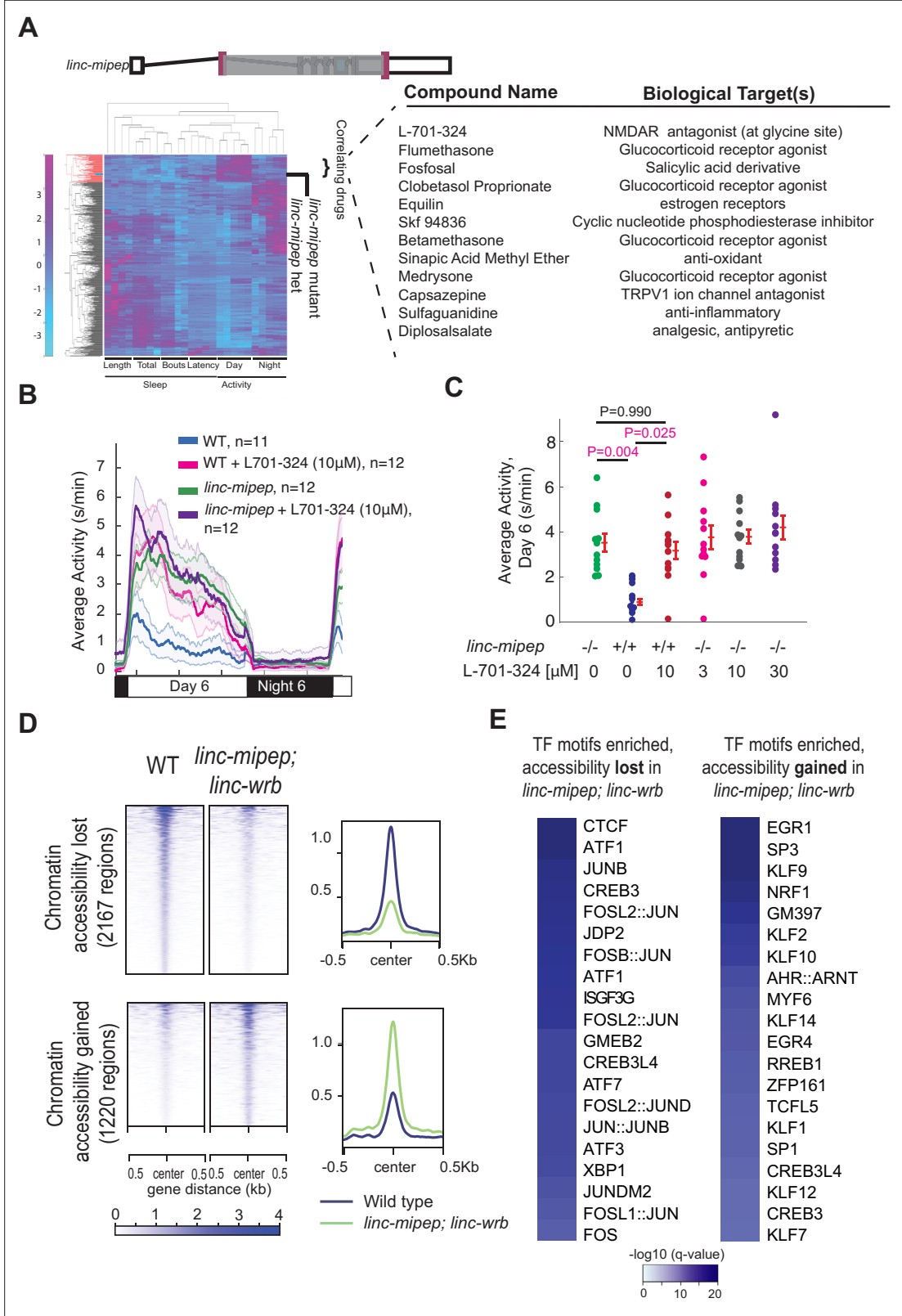

**Figure 3.** *linc-mipep* mutants have dysregulation of NMDA receptor-mediated signaling and immediate early gene induction. (**A**) Left, hierarchical clustering of the *linc-mipep* $^{del-1.8kb}$ (schematic of mutation at top) behavioral fingerprints (right), compared with the fingerprints of wild-type zebrafish larvae exposed to 550 psychoactive agents from 4 to 6 dpf (**Rihel et al., 2010**). The Z score, defined as the average value (in standard deviations) relative to the behavioral profiles of WT exposed to DMSO, is represented by each rectangle in the clustergram (magenta, higher than DMSO; cyan,

*Figure 3 continued on next page*

*Figure 3 continued*

lower than DMSO). The *linc-mipep* <sup>del-1.8kb</sup> fingerprint correlates with agents that induce daytime activity ("Correlating Drugs"). Right, compounds ranked according to correlation with the *linc-mipep* <sup>del-1.8kb</sup> fingerprint, with biological target(s) noted in last column. (**B**) Locomotor average activity of wild-type larvae treated with DMSO (WT, blue) or with 10μM NMDA receptor antagonist L-701,324 (magenta), and *linc-mipep* <sup>del-1.8kb/del-1.8kb</sup> larvae treated with DMSO (*linc-mipep*, green) or with 10μM L-701,324 (purple); sibling-matched larvae tracked over 24hr. (**C**) Average activity (day 6) of WT larvae treated with DMSO or 10μM L-701-324, compared to *linc-mipep*<sup>del-1.8kb/del-1.8kb</sup> larvae treated with DMSO or 3μM, 10μM, or 30μM L-701-324. Each dot represents one fish. L-701–324 has a strong effect in the wild type animals but not in the mutants ($P$=0.05, DrugXGenotype interaction, two-way ANOVA). Key p-values are shown based on Tukey's post-hoc testing. (**D**) Heatmaps (left) and density plots (right) showing chromatin accessibility (omni-ATAC-seq, average of three replicates) profiles of 2167 regions globally with lower accessibility in *linc-mipep; linc-wrb* mutant brains at 5 dpf compared to wild type (WT) brains (top), or 1220 regions globally with higher accessibility in *linc-mipep; linc-wrb* mutant brains at 5 dpf compared to wild type (WT) brains. Heatmaps are centered at the summit of the Omni-ATAC peak with 500bp on both sides and ranked according to global accessibility levels in WT. (**E**) Transcription factor (TF) motifs enriched in up-regulated and down-regulated regions (in **D**), relative to unaffected regions (in *Figure 3—figure supplement 3*).

The online version of this article includes the following figure supplement(s) for figure 3:

**Figure supplement 1.** Correlating small molecules from hierarchical clustering of *linc-mipep* or *linc-wrb* mutant fingerprints with those of wild-type zebrafish larvae exposed to 550 psychoactive agents.

**Figure supplement 2.** *linc-mipep* and *linc-wrb* mutants are sensitized to glucocorticoid receptor agonists.

**Figure supplement 3.** Chromatin accessibility of wild type and *linc-mipep; linc-wrb* mutant brains.

altered accessibility for TF binding sites, which modify the expression of genes involved in neural activation.

## Evolutionarily newer vertebrate brain cell types are more susceptible to loss of *linc-mipep* and *linc-wrb*

Our molecular analyses of wild-type and mutant brains point to gene regulatory networks involved in global transcription rather than neural cell type-specific TFs. We hypothesize that the observed hyper-activity may instead be a result of defects in cells most susceptible to loss of *linc-mipep* and *linc-wrb*. To test this hypothesis, we used single-cell multiomics (transcriptomic and chromatin accessibility) and determined how single cell states are affected in mutant brains compared to sibling-matched WT brains at 6 dpf (*Figure 4a*). To circumvent batch effects from unmatched (non-sibling) samples that may skew single-cell analyses, and because our results so far indicated generally overlapping functions for *linc-mipep* and *linc-wrb*, we chose to analyze *linc-mipep* mutant brain cells and then to validate findings in vivo in *linc-mipep; linc-wrb* double mutants.

First, we used Weighted Nearest Neighbors (WNN) (*Hao et al., 2021*) on transcriptomic and chromatin accessibility data from both *linc-mipep* mutant and WT nuclei all pooled together. This analysis identified 43 clusters (*Figure 4A*, *Figure 4—figure supplement 1A–C*; *Supplementary file 2*). linc-mipep transcripts were detected in all WT clusters except microglia, with a slight enrichment in Purkinje cells, the inhibitory projection neurons of the cerebellum (*Figure 4—figure supplement 1D, E*), raising the possibility that these cells may be more affected in *linc-mipep* mutant brains. *linc-wrb* was detected in all WT clusters except cranial ganglia and ventral habenula cells, and was broadly expressed at lower levels than *linc-mipep* transcripts (*Figure 4—figure supplement 1D, E*). Each cluster was comprised of both WT and *linc-mipep* mutant cells, indicating that there was no complete absence of any cell type in mutants. In *linc-mipep* mutant cells, we note that an almost complete loss of expression of *linc-mipep* was observed in all clusters, without major changes in *linc-wrb* levels (*Figure 3—figure supplement 2A*).

Next, to identify the brain cells most significantly affected by *linc-mipep* mutations, we used Multi-scale PHATE/Integrated Diffusion (*Figure 4B and C*, *Figure 4—figure supplement 3A, B*; *Kuchroo et al., 2022*; *Kuchroo et al., 2021*). This approach measures the effect of *linc-mipep* loss on cellular states by calculating the relative likelihood that any sampled cell state would be observed in either WT or mutant cells. When we analyzed the 'Mutant Likelihood Score' (from *Figure 4B*) for each cell by its respective cluster, we found that differentiating neuronal progenitor, glial progenitor, and cere-bellar granule (excitatory) cell states were more likely to be represented in WT brains, while oligo-dendrocyte progenitor cell (OPC) states were more likely to be represented in *linc-mipep*<sup>-/-</sup> mutant brains (*Figure 4C*, asterisks; *Figure 4—figure supplement 3A, B*; *Supplementary file 6*). Indeed, we find a subcluster of oligodendrocyte progenitor cell states much more likely to be found in

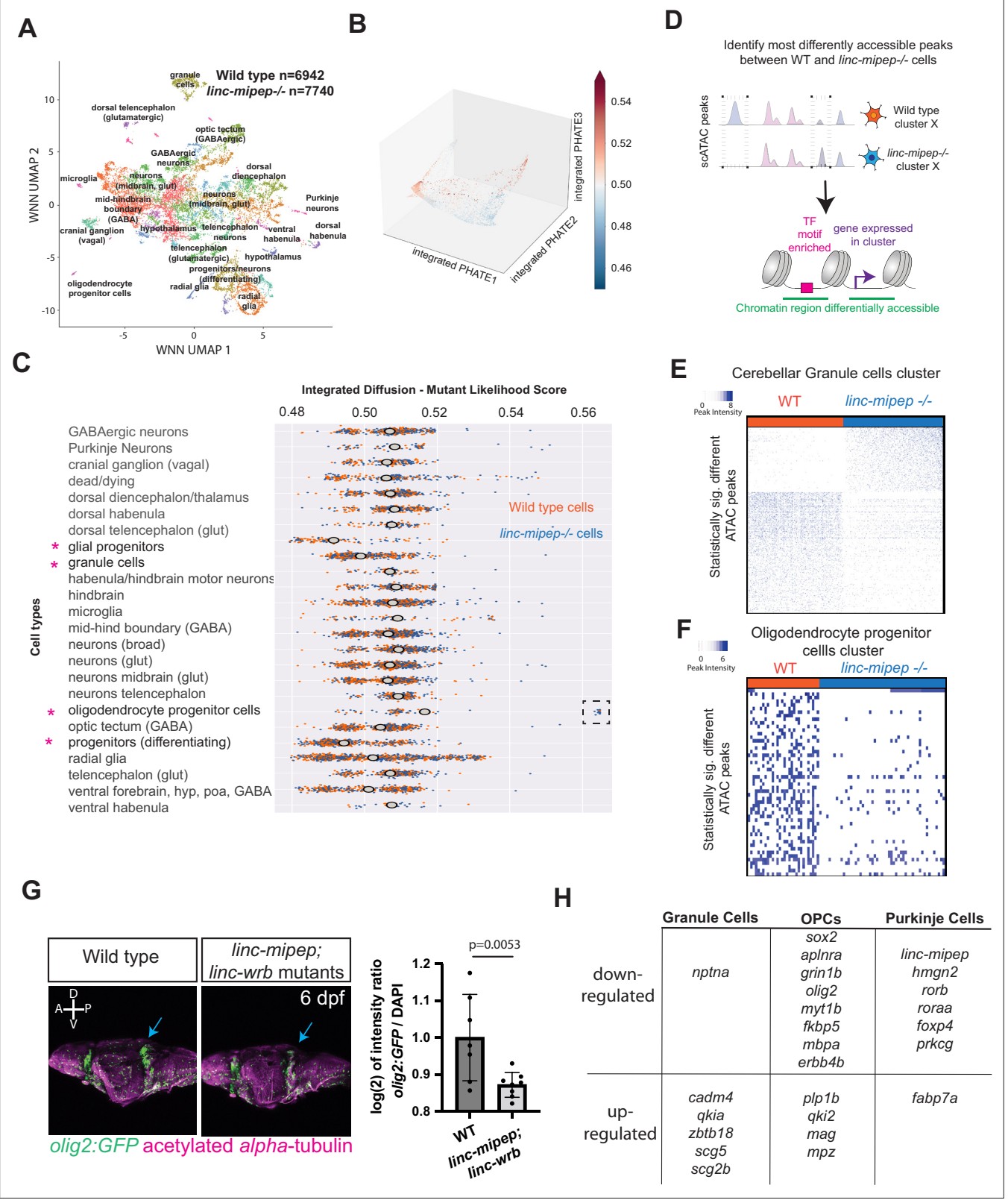

**Figure 4.** Evolutionarily newer vertebrate cell types are more susceptible to loss of *linc-mipep* and *linc-wrb* proteins. (**A**) UMAP representation of WNN analyses of wild type (n=6,942 nuclei) and *linc-mipep* del-1.8kb/del-1.8kb (n=7740 nuclei) mutant brains at 6 dpf. Identified cell types as labeled. (**B**) PHATE plot of integrated diffusion analysis of 6 dpf *linc-mipep* del-1.8kb/del-1.8kb mutant or WT sibling brain nuclei, color-coded by mutant likelihood score as computed by MELD using Integrated Diffusion operator. (**C**) Integrated diffusion analysis on identified cell types from 6 dpf wild type (orange) and *linc-mipep*

*Figure 4 continued on next page*

*Figure 4 continued*

del-1.8kb/del-1.8kb (blue) brains. Each dot represents a single cell, with mutant likelihood score across X-axis. Most wild type- or mutant-like groups noted with an asterisk. Cell types are clustered by known marker genes as defined in **Supplementary file 6**. (**D**) Schematic of analysis to identify most differentially accessible peaks between WT and *linc-mipep* del-1.8kb/del-1.8kb mutant brain nuclei from merged Weighted Nearest Neighbors (WNN) clusters. The most statistically significant changes in chromatin accessibility peaks were identified by the Wilcoxon rank sum and the Kolmogorov-Smirnov (KS) one-tailed tests methods on intensity distributions of each peak in WT and mutant samples, for either wild type or mutant differentially expressed genes per cluster, and for transcription factor (TF) motif overrepresentation by genotype in each cluster.(**E**) Statistically significantly different chromatin accessibility peaks between 6 dpf wild type (WT, blue) and *linc-mipep* del-1.8kb/del-1.8kb mutant (red) nuclei in the cerebellar granule cells cluster. Each column is one nucleus. Color scale, peak intensity (blue, more accessible). (**F**) Statistically significantly different chromatin accessibility peaks between 6 dpf wild type (WT, blue) and *linc-mipep* del-1.8kb/del-1.8kb mutant (red) nuclei in the oligodendrocyte progenitor cells (OPCs) cluster. Color scale, peak intensity (blue, more accessible). (**G**) Left, lateral view confocal images (Z-stack) from *Tg(olig2:GFP)* brains in wild type (left) or *linc-mipep; linc-wrb* double mutant (right) backgrounds at 6 dpf, stained with GFP (*olig2+*, green) and acetylated alpha-tubulin (magenta). A, anterior; P, posterior; D, dorsal; V, ventral. Right, quantification of intensity ratio of GFP+/DAPI signal of whole brain normalized to WT. One-tailed t-test, *P*=0.0053. (**H**) Select differentially regulated genes, down- or up-regulated per each cerebellar granule cells, OPCs, or Purkinje cells cluster. Full list of genes is presented in **Supplementary file 5**.

The online version of this article includes the following figure supplement(s) for figure 4:

**Figure supplement 1.** Single cell Multiome analyses in wild type and *linc-mipep* mutant brain nuclei.

**Figure supplement 2.** *linc-mipep* and *linc-wrb* expression by cluster in wild type or *linc-mipep* mutant brain cells.

**Figure supplement 3.** Single-cell Multiome analyses reveal cell states altered in *linc-mipep* brain cells.

**Figure supplement 4.** Linc-mipep and Linc-wrb protein expression in cerebellar region of *olig2:GFP* brains.

**Figure supplement 5.** Accessibility for transcription factor motifs most affected in *linc-mipep* brain cells.

**Figure supplement 6.** Sample omni-ATAC peaks.

**Figure supplement 7.** dot plot for relevant genes differentially expressed in clusters of interest.

**Figure supplement 8.** Violin plots for NMDA receptor subunits differentially expressed in OPCs.

**Figure supplement 9.** Violin plots for genes differentially expressed in OPCs that are also enriched in granule cells.

*linc-mipep*−/− samples (**Figure 4C**, dashed box). These data indicate that *linc-mipep* preferentially regulates oligodendrocyte and cerebellar cell states during development. Interestingly, these cell states correspond to evolutionarily newer vertebrate brain cell types (**Lamanna et al., 2022**).

In wild-type brains, Linc-mipep and Linc-wrb proteins are expressed throughout the brain. Linc-wrb antibody staining reveals an even expression pattern across the brain at 5 dpf, including in the cerebellar region and in *olig2:GFP+* OPCs (**Figure 4—figure supplement 4C, H–K**). We found that Linc-mipep is more weakly expressed in the torus longitudinalis and tegmentum (as in **Figure 4—figure supplement 4D–G**) compared to Linc-wrb staining. These data suggest that both Linc-mipep and Linc-wrb are expressed in *olig2 +* cells and throughout the cerebellum.

To determine why the mutation affected these particular cell types, we next asked whether loss of *linc-mipep* caused any significant changes in chromatin accessibility and gene expression in single cell types (**Figure 4D**; **Supplementary file 5**). While all cell types are present in both mutant and wild-type brains, we found a strong dysregulation of chromatin accessibility and gene expression within multiple cell types (**Figure 4—figure supplement 3C–E**; **Supplementary file 5**; **Supplementary file 8**). When we examined TF motifs in regions of differential chromatin accessibility in each cluster, we found that *linc-mipep* regulates accessibility for key neurodevelopmental transcription factor families, including Sox, Stat, and Zic family members, in radial glial cells (clusters 3 and 7) and other cells; Esrra/b in midbrain glutamatergic neurons (cluster 13); and Egr and NeuroD members across various cell types (**Figure 4—figure supplement 5A**).

We then examined each cell type of interest more closely to better define cell type-specific changes, starting with cerebellar cell types. In cerebellar granule (excitatory) cells, we found 989 regions where chromatin accessibility was strongly dependent on *linc-mipep* function (645 regions with decreased accessibility, and 344 regions with increased accessibility) (**Figure 4E**). We specifically assessed *linc-mipep* granule cells and found that they lost accessibility at motifs known to bind the transcription factors Bhlhe22, Hic1, which is expressed in mature cerebellar granule cells and transcriptionally represses *Atoh1* (**Briggs et al., 2008**), Neurod2, which required for survival of granule cells (**Miyata et al., 1999**), and Nfia, and gained accessibility at binding sites for Gfi1b and Nfatc1 (**Figure 4—figure supplement 5B**). Purkinje (inhibitory) neurons also showed significant differences in chromatin accessibility (**Figure 4—figure supplement 3E**), losing accessibility at motifs known to

bind the transcription factor Gbx2, which is required for cerebellar development (*Figure 4—figure supplement 5C*; *Wassarman et al., 1997*). Furthermore, examining single-cell expression data, we found that, compared to wild-type Purkinje cells, *linc-mipep* mutant Purkinje cells exhibited a significant decrease in the expression of numerous genes, including *roraa, rorb, foxp4*, and *prkcg*, which are required for maturation or maintenance of Purkinje cells in zebrafish (*Figure 4H*; *Supplementary file 5*; *Takeuchi et al., 2017*). Consistent with these results showing cerebellar cell types are affected, Pol II ChIP-seq in 5 dpf brains showed that genes involved in cerebellar development, including *zic2a, ascl1b*, and *atxn3*, have reduced RNA Polymerase II binding in mutant brains (*Supplementary file 7*).

We next asked whether loss of *linc-mipep* caused any significant changes in chromatin accessibility and gene expression in OPCs. Like cerebellar granule and Purkinje cells, OPCs similarly showed a broad loss of chromatin accessibility in the absence of *linc-mipep* (*Figure 4F*). OPCs from *linc-mipep* mutants showed reduced accessibility at binding sites of E2f7, Elf1 (which is upregulated in differentiating oligodendrocytes), Fev, and Hinfp TFs, and increased accessibility at Sox10 binding sites (*Figure 4—figure supplement 5D*). These changes in accessibility were associated with shifts in gene expression levels consistent with defects in OPC development or maturation, as we found 136 genes that were down-regulated and 57 genes that were up-regulated in *linc-mipep* mutant OPCs relative to wild-type OPCs (*Supplementary file 6*).

To better understand how OPCs may be affected, we further analyzed omni-ATAC-seq analyses in wild type and *linc-mipep;linc-wrb* mutant brains. These analyses revealed differentially accessible regions downstream of *olig2* (a transcription factor that activates the expression of myelin-associated genes), within a large intronic span of *sgms2b* (which synthesizes a component of myelin sheath), and upstream of *fabp7a* (which is important for OPC differentiation in vitro in mouse) (*Foerster et al., 2020*; *Figure 4—figure supplement 6A–C*). To validate that OPCs are affected in vivo, we found a significant 13% decrease (p=0.0053) in *olig2* + oligodendrocyte progenitor cells' signal in mutant brains compared to WT brains, with most of the loss coming from the optic tectum and the cerebellum of *Tg(olig2:eGFP); linc-mipep$^{-/-}$; linc-wrb$^{-/-}$* compared to control larvae (*Figure 4G*; *Figure 4—figure supplement 4A*).

Finally, we asked whether some of the genes that were differentially regulated in OPC, cerebellar granule cell, or Purkinje cell clusters could explain the dysregulation of NMDA receptor signaling and sensitization to glucocorticoids that we found in our earlier pharmacological profiling (*Figure 3*). Indeed, we found that some of the differentially regulated genes in single-cell analyses are known to be involved in NMDA receptor and glucocorticoid receptor signaling. For example, *fkbp5*, which is associated with glucocorticoid signaling, showed reduced expression in *linc-mipep* mutant OPCs, and *scg5*, which can mediate stress responses (*Cao-Lei et al., 2014*; *Mbikay et al., 2001*), showed reduced expression broadly (*Figure 4H*; *Figure 4—figure supplement 7*). These genes also showed changed chromatin accessibility in the *linc-mipep;linc-wrb* double mutant brains (*Figure 4—figure supplement 6D* and E). Similarly, numerous genes involved in NMDA receptor activity (*aldocb, ttyh3b, slc1a2b, nrxn1a, grin1b, gpmbaa, atp1a1b*) showed reduced expression in *linc-mipep* mutant OPCs relative to wild-type OPCs, consistent with a reduction in NMDA receptor signaling in mutants (adjusted p-value = 0.0294, GO Molecular Function from FishEnrichR analysis) (*Supplementary file 6*). For one of these genes, *grin1b*, we also observed associated changes in chromatin accessibility (*Figure 4—figure supplement 6F*). At the single-cell level, we find that the expression of *grin1a* and *grin1b*, which encode NMDAR subunits, are significantly downregulated in *linc-mipep* mutant OPCs relative to wild-type OPCs (*Figure 4H*; *Figure 4—figure supplement 7* and *Figure 4—figure supplement 8A, B*). Some genes that are significantly misregulated specifically in *linc-mipep* mutant OPCs, such as *erbb4, mag, qkia*, and *myt1b*, are also specifically enriched in wild type granule cells, despite different developmental lineage origins and cellular progressions (*Figure 4—figure supplement 9A–D*). Together, these results suggest that loss of *linc-mipep* and *linc-wrb* preferentially affect the development of oligodendrocyte progenitor cells and cerebellar cells – evolutionarily newer vertebrate cell types - and these effects may mediate changes in NMDAR and glucocorticoid signaling through changes in chromatin accessibility and gene expression.

## Discussion

Here, we present the first zebrafish brain single-cell multiome analysis to understand the cell type-specific effects of loss-of-function of the proteins encoded by *linc-mipep* and *linc-wrb*. We found that

mutations in these genes preferentially regulate cerebellar cell types and OPCs and regulate behavior in a dose-dependent manner.

LincRNAs represent a prevalent and functionally diverse class of non-coding transcripts that likely emerged from previously untranscribed DNA sequences, either by duplication from other ncRNAs or from changes of coding regions (*Ulitsky et al., 2011*). Here, we establish that *linc-mipep* (or *lnc-rps25*) and *linc-wrb*, previously identified as long non-coding RNAs, encode micropeptides with homology to the vertebrate-specific non-histone chromosomal protein HMGN1. While it is possible that *linc-mipep, linc-wrb,* and HMGN1 arose from an originally non-coding transcript, possibly in invertebrates, we identify a basal-most vertebrate sequence in lamprey for an ancestral HMGN protein lacking the key C-terminal regulatory domain of human HMGN1. We propose that this ancestral protein may be derived from an unannotated ORF in the invertebrate, basal chordate *Amphioxus* (lancelet) encoding for an APEX1-like gene in the HMGN1 syntenic region. The emergence of *linc-mipep, linc-wrb,* and HMGN1 in jawed vertebrates, and their effects in cerebellar and oligodendrocyte cells, is intriguing. Neural crest cells, myelinating cells (both oligodendrocytes in the CNS and neural crest-derived Schwann cells in the peripheral nervous system), and cerebellar cells (including granule and Purkinje cells) are considered to be among these jawed vertebrate-specific innovations (*Gans and Northcutt, 1983*; *Lamanna et al., 2022*; *Sugahara et al., 2021*). We hypothesize that *linc-mipep, linc-wrb,* and HMGN1 co-evolved with the gene regulatory networks that establish these cell types in development, in line with findings from previous reports (*Deng et al., 2017*; *González-Romero et al., 2015*; *Hock et al., 2007*; *Ihewulezi and Saint-Jeannet, 2021*; *Zalc, 2016*; *Zalc et al., 2008*), as we find that these evolutionarily newer brain cell types are most affected by loss of *linc-mipep* and *linc-wrb* in zebrafish. It will be important for future studies to investigate the effects of the acquisition and evolution of HMGN genes and their preferential roles in the development of these vertebrate cell types (*Deng et al., 2017*; *González-Romero et al., 2015*; *Hock et al., 2007*; *Ihewulezi and Saint-Jeannet, 2021*; *Zalc, 2016*; *Zalc et al., 2008*).

We find that mutations in *linc-mipep* and *linc-wrb* most affect cerebellar granule and Purkinje cells and OPCs and behavior. Both OPCs and cerebellar cells are typically associated with post-natal growth in humans. The cerebellum is a folded hindbrain structure important for coordinating body movements and higher-order cognitive functions. Our results suggest a plausible explanation for with recent findings in Trisomy 21 (Down syndrome) pathology, in which HMGN1 is overexpressed, that developing and adult Down syndrome brains have dysregulated expression of genes associated with oligodendrocyte development and myelination in addition to alterations in the cerebellar cortex (*Baxter et al., 2000*; *Mowery et al., 2018*; *Olmos-Serrano et al., 2016*), highlighting the important roles that oligodendrocytes play in normal neurodevelopment and neurodevelopmental disorders (*Jin et al., 2020*). Our behavioral mutant analyses highlight the dose-dependent roles of *linc-mipep* and *linc-wrb;* evolutionarily conserved functions between *linc-mipep, linc-wrb,* and human HMGN1 in neurodevelopment (*Abuhatzira et al., 2011*; *Deng et al., 2017*; *Deng et al., 2013*); and the importance of understanding the ancestral and conserved roles of key neurodevelopmental genes in non-mammalian and more basal vertebrate systems. Altogether, these studies emphasize the importance of non-neuronal and non-cerebral cortex cell types in neurodevelopmental disorders (*Sathyanesan et al., 2019*), in which the vertebrate-specific *Hmgn1* and related proteins may play a unifying role by regulating chromatin accessibility for key transcription factors.

Our results indicate that loss of *linc-mipep* and *linc-wrb* has an effect on chromatin accessibility, which has an effect on the regulatory activity of multiple TFs and gene expression networks. In particular, chromatin accessibility in mutants is altered at *grin1b,* among other regions, and we find differential regulation of other genes in *linc-mipep* mutant OPCs related with NMDA receptor signaling. These data provide a potential mechanism for how these genes are significantly differentially expressed between wild type and *linc-mipep* mutant OPCs. However, future studies will be needed to understand how these non-histone chromosomal proteins regulate not only this pathway but other epigenetic aspects of neural development and cell function. One possibility is that NMDA signaling is preferentially dysregulated in these cells. Alternatively, NMDA signaling may be broadly dysregulated, while affecting these cells the most. Evidence from early mouse development found that NMDA receptors are most abundant in oligodendrocyte progenitor cells compared to mature oligodendrocytes (*De Biase et al., 2010*; *Zhang et al., 2014*). One study proposes that a main role specifically for NR1 (encoded by *Grin1* in mouse) is to maintain oligodendrocyte glucose transport, which is crucial

for the function and health of myelinated axons (*Saab et al., 2016*). Future investigations will have to reveal exactly how loss of these zebrafish HMGN1 homologs affects the development and maintenance of oligodendrocytes and cerebellar cells and how the intricate cross-talk between these cells is affected in *linc-mipep* and *linc-wrb* mutants. It will also be important to define how the proteins encoded by *linc-mipep* and *linc-wrb* specifically regulate NMDAR signaling and whether this mechanism is conserved in other vertebrate species. Some studies of HMGN1 in mammalian cells have elucidated some of its key molecular mechanisms of gene regulation (*Deng et al., 2017*; *Abuhatzira et al., 2011*; *He et al., 2018*; *Prymakowska-Bosak et al., 2002*; *Catez et al., 2002*; *Lim et al., 2005*). Future work will be needed to fully uncover the molecular mechanisms and binding/interaction partners for each protein in zebrafish and across other vertebrate species, to understand to what extent these mechanisms are conserved. We also do not know whether these paralogous genes work cooperatively or redundantly. For example, future work should investigate whether these related genes have distinct and/or partially overlapping targets and binding partners.

Finally, screening for behavioral phenotypes using F0 mutants is emerging as an important way to decrease time and number of vertebrate animals to enrich for gene candidates for further study (*Kroll et al., 2021*). Further advances have also increased the resolution of behavioral parameters or patterns affected, allowing for more detailed phenotyping and downstream analyses (*Kroll et al., 2021*; *Ghosh and Rihel, 2020*). This phenotyping approach can further enable screens for other micropeptides that are identified through ribosome profiling or mass spectrometry, lincRNAs, and rare or unannotated candidate disorder risk genes. We note limitations for targeting some of these genes are lower GC content, shorter exon lengths, and inducing larger deletions that may cause a phenotype as a result of a necessary noncoding element. However, there are now non-canonical Cas9s and Cas13s and nearly-PAMless endonucleases that can be tested (*Treichel and Bazzini, 2022*; *Vicencio et al., 2022*). Current efforts in the field are underway to understand how F0 phenotyping is similar or different from phenotypes observed in stable mutants. Nonetheless, mutations such as those presented in *Figure 1D* will be important to decipher the role(s) of micropeptides or lincRNAs, including some genes that may have multiple coding and non-coding functions.

Overall, this study highlights the power of using a high-throughput, genetically tractable vertebrate model to systematically screen for micropeptide function within putative lincRNAs, behavioral phenotypes, signaling pathways, and cell type susceptibilities in early vertebrate development. How novel protein-coding genes may be born from non-coding genomic elements remains an elusive question (*Weisman, 2022*). Several short open reading frames encoding for functional, evolutionarily conserved peptides now have been discovered within putative non-coding RNAs (*Makarewich and Olson, 2017*), and some of these genes may have emerged along vertebrate lineages (for example, *libra*/NREP *Bitetti et al., 2018*). Our analyses support the idea that many more unannotated or undescribed proteins may similarly play critical roles in vertebrate neurodevelopment and behavior (*Barlow et al., 2020*). We propose that revisiting sORFs identified within putative long non-coding RNAs in basal vertebrates may provide insight into gene innovation and evolution. This framework will enable genetic studies in a basal system to understand the evolutionary origins of human developmental disorders and diseases in a vertebrate cell type-specific manner.

# gMaterials and methods
## Zebrafish husbandry and care

Fish lines were maintained in accordance with the AAALAC research guidelines, under a protocol approved by the Yale University Institutional Animal Care and Use Committee (IACUC Protocol Number 2021–11109). We have complied with all relevant ethical regulations under this protocol. Zebrafish husbandry and manipulation were performed as described, and all experiments were carried out at 28 °C. For all larval experiments, zebrafish embryos were raised at 28.5 °C in petri dishes at densities of 70 embryos/dish on a 14 hr:10 hr light:dark cycle in a DigiTherm 38 liter Heating/Cooling Incubator with circadian lighting (Tritech Research). Dishes of embryos were cleaned once per day with blue water (fish system water with 1 mg/L methylene blue, pH 7.0) until they were placed in behavior boxes (ZebraBox, Viewpoint), to ensure identical growing conditions. Normal development was assessed, and larvae exhibiting abnormal developmental features (no inflated swim bladder, curved) were not used.

## Ribo-seq profiles

Sequences for ribosome profiling were previously published (*Bazzini et al., 2014* and *Johnstone et al., 2016*). Code for updated ribosome profiling plots available here. Updated mapping, including for new genome releases, is available here.

## CRISPR F0 experiments

Synthetic guides were designed using CRISPRscan and ordered as sgRNAs through Synthego (Synthego Corportation, Redwood City, CA, USA). Target and scrambled (control) sequences are presented in *Supplementary file 1*. EnGen Spy Cas9 NLS protein (NEB, M0646) was used for F0 experiments. RNPs were formed by mixing 3 µM Cas9 protein, 300 mM KCl, and 10 mM of each synthetic sgRNA targeting one gene, incubating at 37 °C for 10 min, and cooling to room temperature for 5 min. One-cell stage zebrafish embryos were injected with 100pl of each respective mix early after fertilization into the yolk. Pools of 8 embryos at 24 hpf were collected and incubated in 50 µl of 100 mM NaOH at 95 °C for 20 min. Then, 25 µl of 1 M Tris-HCl (pH 7.5) was added to neutralize the mix. Two µl of these crude DNA extracts were used for genotyping with the corresponding forward and reverse primers (10 µM; *Supplementary file 1*) using a standard PCR protocol, and these products were then sent for Sanger sequencing to assess cutting efficiencies. Mutation efficiency was assessed using Inference of CRISPR Edits (Synthego Performance Analysis, ICE Analysis. 2019. v3.0. Synthego). We note that the *linc-mettl3* target sites lie between highly repetitive regions, making it difficult to amplify the necessary length for ICE analysis. We provide PCR and Sanger sequencing results in this case, indicating efficient targeting and significant large genomic deletion.

## CRISPR mutant generation

CRISPR mutant generation was done following *Moreno-Mateos et al., 2015*. Briefly, CRISPRScan ( crisprscan.org) was queried to identify appropriate target sequences (*Moreno-Mateos et al., 2015*). Primers were ordered and amplified with universal primer 5'- AAAAGCACCGACTCGGTGCCACTT TTTCAAGTTGATAACGGACTAGCCTTATTTTAACTTGCTATTTCTAGCTCTAAAAC-3'. sgRNAs were in vitro transcribed using the AmpliScribe T7 Flash kit, using the PCR product (with T7 promoter) as template. In vitro transcribed sgRNAs were treated with DNase I and precipitated with sodium acetate and ethanol. *Cas9* mRNA was in vitro transcribed from DNA linearized by XbaI (pT3TS-nCas9n) using the mMESSAGE mMACHINE T3 kit (Ambion). In vitro transcribed Cas9 RNA was treated with DNase I and purified using the RNeasy Mini kit (Qiagen).

One-cell stage zebrafish embryos were injected with 50 pg of each respective sgRNA and 100 pg of cas9 mRNA. sgRNA and genotyping primers and target sequences are available in *Supplementary file 1*.

## Overexpression constructs

gBlocks (IDT) were ordered for the *linc-mipep* or human *Hmgn1* coding sequence, plus a FLAG and HA tag at the C terminus, as follows:

> *Linc-mipep CDS:* 5'-gccaccATGCCTAAAAGGAGCAAAGCGAACAATGACGCT GAAGTCTC TGAGCCTAAAAGAAGGTCAGAGAGGTTGGTAAACAAACCTGCACCCCCAAAGGCAGAGCC CAAGCCAAAGAAGGCCCCTGCCAAACCTAAGAAAACAAAGGAACCCAAGGAGCCCA AGGAGGAGGAGAAGAAAGAGGAGGTGCCCGCAGAAAACGGAGAAACAAAAGCTGAC GATGATGCATCGGCAACAGAAGACGGCGACAAGAAAGAAGACGGGGAAGGTTCTGG CTCAgactacaaagacgatgacgacaagtacccatacgatgttccagattacgctTAA-3'
> Human Hmgn1CDS: 5'-gccaccATGCCCAAGAGGAAGGTCAGCTCCGCCGAAGGCGCCGC CAAGGA
> AGAGCCCAAGAGGAGATCGGCGCGGGTTGTCAGCTAAACCTCCTGCAAAAGTGGAAG CGAAGCCGAAAAAGGCAGCAGCGAAGGATAAATCTTCAGACAAAAAAGTGCAAACA AAAGGGAAAAGGGGAGCAAAGGGAAAACAGGCCGAAGTGGCTAACCAAGAAACTAA AGAAGACTTACCTGCGGAAAACGGGGAAACGAAGACTGAGGAGAGTCCAGCCTCTG ATGAAGCAGGAGAGAAAGAAGCCAAGTCTGATGGTTCTGGCTCAgactacaaagacgatgacga caagtacccatacgatgttccagattacgctTAA-3'

Addgene plasmid #79885 (pMT-ubb-cytoBirA-2a-mCherry, a gift from Tatjana Sauka-Spengler *Trinh et al., 2017*) was digested with BamHI and EcoRV, and the resulting vector was used as the backbone for the construct. InFusion cloning (Takara Bio) was used to amplify the Linc-mipep-FLAG-HA coding sequence and ligate with the vector, using primers F: 5'- TTGTTTACAGGGATCgccaccATGCCT AAAAGGAGC-3' and R: 5'- CTCTCCTGATCCGATagcgtaatctggaacatcgtatggg-3'. InFusion cloning (Takara Bio) was used to amplify the human Hmgn1-FLAG-HA coding sequence and ligate with the vector, using primers F: 5' TTGTTTACAGGGATCCGCCACCATGCCCAAGAGG –3' and R: 5'-CTCT CCTGATCCGATATCATCAGACTTGGCTTCTTTCTCTCC-3'. Sequence-verified plasmids were midi-prepped and injected into the cell of one-cell stage embryos at 20 ng/µl along with 200 ng/ul of Tol2 transposase capped mRNA.

## Fish lines used in this study

The following stable fish mutant lines have been established in this study: *linc-mipep*$^{del1.8kb}$ (ya126); *linc-mipep*$^{ATG-del6}$ (ya127); *linc-mipep*$^{del8}$ (ya128); *linc-mipep*$^{3'UTR-del74}$ (ya129); *linc-wrb*$^{del11}$ (ya130). The following stable transgenic lines have been established in this study: *Tg(ubb:linc-mipep-FLAG-HA-T2A-mCherry)* (ya145); and *Tg(ubb:human-Hmgn1-FLAG-HA-T2A-mCherry)* (ya151). The following previously published transgenic line has been used in this study: *Tg(olig2:egfp)*$^{vu12}$.

## Quantitative locomotor activity tracking and statistics for sleep/wake analyses

At 4 dpf, single larvae from heterozygous *linc-mipep* mutant incrosses were placed into individual wells of a clear 96-square well flat plate (Whatman) filled with 650 µL of blue water (fish system water with 1 mg/L methylene blue, pH 7.0). Plates were placed in a Zebrabox (ViewPoint Life Sciences), and each well was tracked using ZebraLab (Viewpoint) in quantized mode, and analyzed with custom software as in *Rihel et al., 2010* and at *Rihel, 2023* and DOI: 10.5281/zenodo.7644073. Behavioral data were analyzed for statistical significance using one-way ANOVA followed by Tukey's post hoc test ($\alpha$=0.05), as previously described (*Rihel et al., 2010*). Each behavioral experiment presented was repeated 2–4 times. For analyses of maternal-zygotic *linc-mipep;linc-wrb* mutants, age- and size-matched wild type adult stocks (AB/TL) or *linc-mipep;linc-wrb* double-homozygous mutants were incrossed, collected simultaneously, and raised in identical conditions prior to quantitative locomotor activity tracking as described above.

## Behavioural fingerprints and Euclidean distances

As previously described (*Kroll et al., 2021*), the raw file generated by the ZebraLab software (ViewPoint Life Sciences) was exported into a series of xls files each containing 1 million rows of data. Each datapoint represented the number of pixels that changed grey value above a sensitivity threshold, set to 18, for one larva at one frame transition, a metric termed Δ pixels. These files, together with a metadata file labelling each well with a genotype, were input to the MATLAB script Vp_Extract.m (*Ghosh and Rihel, 2020*), which calculated the following behavioral parameters from the Δ pixels timeseries for both day and night: (1) active bout length (duration of each active bout in seconds); (2) active bout mean (mean of the Δ pixels composing each active bout); (3) active bout standard deviation (mean of the Δ pixels composing each active bout); (4) active bout total (sum of the Δ pixels composing each active bout); (5) active bout minimum (smallest Δ pixels of each bout); (6) active bout maximum (largest Δ pixels of each bout); (7) number of active bouts during the entire day or night; (8) total time active (% of the day or night); (9) inactive bout length (duration of each pause between active bouts in seconds). These measurements were then averaged across both days or both nights to obtain one measure per parameter per larva for the day and night. To build the behavioral fingerprints, we calculated the deviation (Z-score) of each mutant (F0) larva from the mean of their wild-type siblings across all parameters. Plotted in Extended Data *Figure 1b and c* for each parameter is the mean ± SEM of the Z-scores. We compared fingerprints between replicates (Extended Data *Figure 1b*) or between *linc-mipep* and *linc-wrb* (Extended Data *Figure 1c*) using Pearson correlation. The behavioral fingerprint of each larva can be conceptualized as a single datapoint in a multidimensional space where each dimension represents one behavioral parameter. To summarize the intensity of each phenotype across parameters, we measured the Euclidean distance between each larva and the mean fingerprint of its wild type siblings, set at the origin of this space by the Z-score normalization (*Figure 1—figure*

*supplement 3B*). Generally, F0 mutants with more parameters affected, or with more extreme differences in the parameters affected, displayed larger Euclidian distances; those with few or with mildly-affected parameters displayed smaller Euclidean distances. Code for this analysis is available on GitHub (*Kroll, 2022*). Prism 9 (GraphPad) was used for statistics and plotting for *Figure 1—figure supplement 4*.

## Hierarchical clustering

Correlation analysis was done in MATLAB (R2018a; The MathWorks) as previously described (*Rihel et al., 2010*). Behavioral phenotypes of wild-type fish exposed to a panel of 550 psychoactive agents from 4 to 7 dpf were ascertained as previously described (*Rihel et al., 2010*). To compare the behavioral fingerprints of WT larvae exposed to each drug and the *linc-mipep* mutant behavioral fingerprint, hierarchical clustering analysis was performed as in *Rihel et al., 2010*; *Hoffman et al., 2016*.

## Sequence alignments and homologies

BLASTp, BLASTn, and the UCSC Genome Browser were used to find sequences (especially the highly conserved 3'UTR sequence) and proteins with sequence homology and/or synteny to human *Hmgn1*. Clustal Omega (through EMBL-EBI) was for multiple sequence alignments.

## Custom antibodies generation

Three custom antibodies were designed (YenZym Antibodies, LLC) against: *Si:ch73-1a9.3 (linc-mipep)*, C-Ahx-DDASATEDGDKKEDGE-COOH; *Si:ch73-281n10.2 (linc-wrb)*, C-Ahx-EDAKPEAEEKTP-amide; and both *Si:ch73-1a9.3* and *Si:ch73-281n10.2*: KRSKANNDAE-Ahx-amide. The last antibody designed to recognize both proteins was non-specific and not further used. Antibody specificity was confirmed by antibody staining in wild type and *linc-mipep; linc-wrb* mutants.

## Antibody staining and imaging

Embryos up to 24 hpf: Embryos were dechorionated and collected into room-temperature 4% PFA in PBS for 1 hr. Embryos were blocked rotating for 1 hr at room temperature in 10% normal goat serum (NGS) (Thermo Fisher Scientific, 50062Z), primary antibody stained for 1 hr at room temperature in 10% NGS, washed 3x5 min in 1xPBS with 0.25% Triton-X (PBST), incubated rotating and protected from light for 1 hr at room temperature, washed 3x5 min in PBST, and mounted in 0.7–1% low-melt agarose on glass-bottom dishes (MatTek) for imaging. *Larvae*: Larvae (up to 6 dpf) were maintained in a quiet environment. For assessment of *olig2* + cells, the *Tg(olig2:egfp)$^{vu12}$* line (*Shin et al., 2003*) was crossed to either wild type or double-homozygous *linc-mipep;linc-wrb* mutants. Subsequently, those *olig2:egfp* adults were outcrossed to either wild type or *linc-mipep1;linc-wrb* double homozygous mutants. To ensure rapid fixation at 6 dpf, larvae from each of these crosses were poured through a mesh sieve and immediately submerged into ice-cold 4% PFA (Electron Microscopy Sciences) /1 x PBS-0.25% Triton X-100 (PBST)/4% sucrose, in fix, as previously described (*Randlett et al., 2015*). Larvae were fixed overnight at 4 °C and washed three times for 15 min each in PBST. For whole larvae, pigment was bleached with a 1% $H_2O_2$/3% KOH solution (in PBS), washed 3x15 min in PBST, then permeabilized with acetone (pre-cooled to –20 °C) at –20 °C for 20 min, and washed three times for 15 min with PBST. For dissecting brains (critical for assessment of GFP + cells), following overnight fixation, larvae were washed 3x5 min in PBST, then brains were dissected by hand and transferred back into tubes with PBS. Brains were sequentially dehydrated 5 min each in 25% MeOH/75% PBS, 50% MeOH/50% PBS, 75% MeOH/50% PBS, and 100% MeOH, and stored at –20 °C for at least overnight. Brains were sequentially similarly rehydrated, then permeabilized with 1 x Proteinase K (10 mg/ml is 1000 x stock) in PBST for exactly 10 min. Brains were then rinsed 3 x with PBST, post-fixed in 4% PFA/PBST for 20 min at room temperature, and washed three times for 5 min in PBST. Samples were mounted in 0.7–1% low-melt agarose on glass-bottom dishes (MatTek) for imaging. Confocal imaging was performed using a Zeiss 980 AiryScan or a Leica SP8 confocal microscope. Images were processed and analyzed using FIJI software and plugins.

## Primary antibodies used

custom Linc-mipep (rabbit); custom Linc-wrb (rabbit); anti-GFP (mouse, A11120, Thermo Fisher Scientific, 1:500); acetylated α-tubulin (rabbit, 5335T, Cell Signaling Technology, 1:500). Alexa Fluor 488,

546 or 568 secondary antibodies against rabbit or mouse were used at 1:500 (Invitrogen). DAPI (for nuclear marking) was added at 1:10,000 during secondary antibody staining.

## RNA in situ hybridization

Template DNAs for antisense RNA probes were amplified from a pool of 6 hpf, 1 dpf, 2 dpf, and 5 dpf zebrafish cDNA using primers containing the T7-promoter sequence in the reverse primer. All sequences are listed in *Supplementary file 1*. Digoxigenin (DIG)-labeled RNA probes were synthesized using T7 RNA Polymerase (Roche) and purified using Monarch RNA Cleanup Kit (New England Biolabs). RNA in situ hybridization was performed as described (*Giraldez et al., 2005*; *Thisse and Thisse, 2008*). Briefly, embryos at the respective stages were dechorionated (if applicable) and fixed with 4% paraformaldehyde (PFA) overnight at 4 °C. Fixed embryos were washed 3 X with 1 x phosphate-buffered saline (PBS), then dehydrated with a methanol series (25%, 50%, 75%, and 100% methanol). Dehydrated embryos were stored in 100% methanol for at least 24 hr at –20 °C. Embryos were then rehydrated with a reverse methanol series and washed with 1 x PBS. Pre-hybridization and hybridization were performed at 65 °C for 3 hr and overnight, respectively. Embryos were washed extensively and blocked for 3 hr at room temperature, then incubated with anti-DIG antibody overnight at 4 °C. After antibody incubation, embryos were stained with BCIP/NBT, and staining was stopped with 4% PFA overnight at 4 °C. Embryos were then washed briefly, mounted with a glycerol series (50%, 70%, and 86%), and imaged in 86% glycerol with a Zeiss stereo Discovery.V12 microscope.

## RNA-seq and qPCR

Data in *Figure 1—figure supplement 5F* was generated using publicly available RNA-sequencing data (*White et al., 2017*). For qPCR, larvae (n=10 per sample) were pooled and flash-frozen in liquid nitrogen and stored at –80 °C. Trizol (Invitrogen) was added to samples and homogenized with sterile pestles. Chloroform was then added, and samples were centrifuged at 4 °C for 15 min at 12,000 x *g*. The aqueous supernatant was placed into a new tube, and isopropanol was added along with 1 µl of GlycoBlue. Samples were left at –20 °C for 2 hours and centrifuged at 4 °C for 15 min at 12,000 x *g*. The pellet was washed two times with RNase-free 70% ice-cold ethanol, dried, and resuspended in RNase-free water. 1 µg of RNA was used to make cDNA with the SuperScript III First-Strand Synthesis system (Invitrogen). cDNA was diluted 1:3, and 1 µl was used for each qPCR sample using Power Sybr Green Master Mix (2 x) and respective primers, in technical triplicates. Primers for amplification: *fosab (c-fos)*, 5'- GTGCAGCACGGCTTCACCGA-3' and 5'- TTGAGCTGCGCCGTTGGAGG-3'; *ef1a1l1*, 5'-TGCTGTGCGTGACATGAGGCAG-3' and 5'-CCGCAACCTTTGGAACGGTGT-3' (*Reichert et al., 2019*). Expression of *fosab (c-fos)* was normalized to the expression of *ef1a1l1 for each respective sample and timepoint,* and relative expression levels were calculated using the ΔΔCt method.

## Western blot

Embryos from wild type or *Tg(ubb:linc-mipep-FLAG-HA-T2A-mCherry)* incrosses were dechorionated at 6 hpf, and 150 embryos were collected per sample per replicate. Water was removed, and embryos were deyolking in 500 µl Deyolking Buffer (55 mM NaCl, 1.8 mM KCl, 1.25 mM NaHCO3) by pipetting through a narrow tip to disrupt the yolk sac. Embryos were shaken at 1100 rpm for 5 min. Cells were then pelleted at 300 *g* for 30 s, and the supernatant was discarded. Two wash steps were performed using wash buffer (110 mM NaCl, 3.5 mM KCl, 2.7 mM CaCl2, 10 mM Tris/Cl pH 8.5), shaking two minutes at 1100 rpm and pelleting cells. The supernatant was then removed and samples were flash-frozen in liquid nitrogen. Cell pellets were then resuspended in 100 µl sample buffer (1 x NuPAGE LDS Sample Buffer supplemented with DTT). After heating for 10 min at 95 °C, protein samples (40 µl, ~60 deyolked embryos) were resolved on a 4–12% Bis-Tris gel with NuPAGE MOPS Running Buffer (Thermo Fisher Scientific) and transferred to a nitrocellulose membrane using the iBlot 2 Gel Transfer Device (Thermo Fisher Scientific). Membranes were blocked in 5% milk / PBS with 0.1% Tween-20 (PBST), incubated with primary antibody solution (each antibody at 1:2000) prepared in block solution, and then incubated with a peroxidase-conjugated secondary antibody solution prepared in block solution. Proteins were detected with SuperSignal West Pico PLUS Chemiluminescent Substrate (for Actin antibody) or SuperSignal West Femto Maximum Sensitivity Substrate (for FLAG antibody; Thermo Fisher Scientific).

## In vivo pharmacological drug experiments

At 4 dpf, single larvae from heterozygous *linc-mipep* mutant incrosses were placed into individual wells of a clear 96-square well flat plate (Whatman) filled with 650 µL of blue water. Respective pharmacological agents (from a stock of 5 or 50 mM depending on solubility) or corresponding vehicle controls (DMSO or water) were pipetted directly into the water to achieve the desired final concentrations at the start of the experiment (typically evening of 4 dpf). Since both *linc-mipep* and *linc-wrb* had similar hyperactivity profiles, we focused on *linc-mipep* to allow for drug analyses of mutant and wild type (WT) larvae with matched genetic backgrounds. Drug treatments, vehicles, and doses are described in *Supplementary file 3*.

## Genotyping

After each behavioral tracking experiment, larvae were anesthetized with an overdose of MS-222 [0.2–0.3 mg/ml], transferred into 96-well PCR plates, and incubated in 50 µl of 100 mM NaOH at 95 °C for 20 min. Then, 25 µl of Tris-HCl 1 M pH 7.5 was added to neutralize the mix. Two µl of these crude DNA extracts were used for genotyping with the corresponding forward and reverse primers (10 µM; *Supplementary file 1*) using a standard PCR protocol.

## Brain collection for molecular analyses

Briefly, brains at peak daytime activity levels (Zeitgeber Time 4, i.e. 4 hr after lights on) were dissected from 5 dpf MZ-*linc-mipep;linc-wrb* or wild type zebrafish (for omni-ATAC-seq n=10 per sample, and ChIP-seq n=50 per sample) or 6 dpf zebrafish from one *linc-mipep*[-/-] heterozygous incross (for single-cell Multiome, n=12 per sample) in ice-cold Neurobasal media supplemented with B-27 (Thermo Fisher Scientific), snap-frozen in a dry ice/methylbutane bath (to preserve nuclear structure), and stored at –80 °C until use. Trunks of *linc-mipep* fish from the heterozygous cross were genotyped, then wild type or *linc-mipep*[-/-] brains as confirmed by genotyping were pooled together before proceeding with scMulitome.

For ChIP-seq experiments, brains were dissected and homogenized before treatment with 1% PFA (protocol adapted from *Cotney and Noonan, 2015*) and performed as previously described *Miao et al., 2022* using 4 µg of RNA Polymerase II antibody (ab817, Abcam) per sample; 5% input samples were also collected and processed.

Omni-ATAC was performed on frozen brains from 5 dpf zebrafish based on published protocols (*Buenrostro et al., 2013*; *Corces et al., 2017*). Frozen brain tissue was homogenized in cold homogenization buffer (320 mM sucrose, 0.1 mM EDTA, 0.1% NP40, 5 mM CaCl2, 3 mM Mg(Ac)2, 10 mM Tris pH 7.8, 1×protease inhibitors (Roche, cOmplete), and 167 µM β-mercaptoethanol) on ice. The lysate was filtered with a tip strainer (Flowmi Cell Strainers, porosity 70 µm) into a new Lo-Bind tube. Nuclei were isolated using the gradient iodixanol solution as described (*Corces et al., 2017*). Nuclei solution was mixed with 1 ml of dilution buffer (10 mM Tris-HCl pH 7.4, 10 mM NaCl, 3 mM MgCl2, 0.1% Tween-20) and was then centrifuged at 500 x *g* for 10 min at 4°C. Transposition and library preparation were performed on the purified nuclei as described (*Miao et al., 2022*).

The supernatant was removed, and the purified nuclei were resuspended in the transposition reaction mixture (25 µl 2×TD Buffer, 2.5 µl Tn5 transposase, 22.5 µl Nuclease-Free water) and incubated for 30 min at 37 °C. DNA was then purified with the Qiagen MinElute Kit (Qiagen, 28004). Libraries were prepared using NEBNext High-Fidelity 2 X PCR Master Mix (NEB, M0541) with the following conditions: 72 °C, 5 min; 98 °C, 30 s; 15 cycles of 98 °C, 10 s; 63 °C, 30 s; and 72 °C, 1 min. Libraries were purified with Agencourt AMPureXP beads (Beckman Coulter Genomics, A63881) and sequenced with the Illumina NovaSeq 6000 System at the Yale Center for Genome Analysis.

## High-throughput sequencing data management

LabxDB seq (*Vejnar and Giraldez, 2020*) was used to manage our high-throughput sequencing data and configure our analysis pipeline. Export to the Sequence Read Archive was achieved using the "export_sra.py" script from LabxDB Python. All sequencing datasets generated in this work have been deposited through NCBI, BioProject PRJNA945049. Detailed information about these datasets are also provided in *Supplementary file 9*.

## Omni-ATAC data processing, differential and motif enrichment analysis

Raw paired-end Omni-ATAC reads were mapped using LabxPipe (*Vejnar, 2023b*). Reads were adapter trimmed using ReadKnead (*Vejnar, 2023d*) and mapped to the zebrafish GRCz11 genome sequence

*Yates et al., 2020* using Bowtie2 (*Langmead and Salzberg, 2012*) with parameters '-X 2000, `--no-unal`, `"--no-unal"`, `"--no-mixed"`, `"--no-discordant"`. The alignments were deduplicated using samtools markdup (*Li et al., 2009*). For genome-wide analysis, only uniquely mapped reads (with alignment quality ≥30) were used. Reads mapped to the + strand were offset by +4 bp and reads mapped to the – strand were offset by −5 bp (*Buenrostro et al., 2013*). Only fragments with insert size ≤ 100 bp (effective fragments) were used to determine accessible regions. Genome tracks were created using BEDTools (*Quinlan and Hall, 2010*) and utilities from the UCSC genome browser (*Lee et al., 2022*). For all the genome tracks in the paper, signal intensity was in RPM (reads per million). Fragment coverage on each nucleotide was normalized to the total number of effective fragments in each sample per million fragments.

## Peak calling

Effective reads from three *linc-mipep;linc-wrb* double mutant replicates and three wild-type replicates were merged. Then narrow peaks were called on the merged data using MACS3 (*Zhang et al., 2008*) with the additional parameters '-f BEDPE `--nomodel` --keep-dup all' with significance cutoff at $p=10^{-20}$. In total, 173,443 narrow peaks were called. Among them, 170,599 peaks were located within chromosomes 1–25; these regions were determined as accessible regions for further analysis. Differential analysis was performed using DESeq2 (*Love et al., 2014*), comparing fragment coverage of each accessible region in the three *linc-mipep;linc-wrb* double mutant replicates with that in the three wild-type replicates. A total of 3367 regions that were mapped to chromosomes 1–25 show a significant difference (false discovery rate (FDR)<0.01), with 2167 regions significantly up-regulated and 1200 regions significantly down-regulated. A total of 2928 unaffected regions (FDR >0.95; 1.005<*linc-mipep;linc-wrb* / WT <0.995) were used as control regions for plotting and motif enrichment analysis. Accessibility heatmaps and density plots were generated using deeptools (*Ramírez et al., 2014*).

## Motif enrichment analysis

This was performed on the up-regulated and the down-regulated regions, with unaffected regions as control, using AME in MEME suite (*McLeay and Bailey, 2010*) with default parameters (https://meme-suite.org/meme/tools/ame, motif database option: Vertebrates In vivo and in silico, Eukaryotic DNA). Motif heatmaps were generated using the R package gplots (*Warnes et al., 2022*). Tracks for omni-ATAC-seq of wild type or *linc-mipep; linc-wrb* mutant brains are publicly available at https://www.giraldezlab.org/data/tornini_et_al_2023_elife/.

## ChIP-seq data processing and analysis

Raw ChIP-seq reads were adapter trimmed, mapped, and deduplicated using the same method described in the previous section but using the default parameters for Bowtie2 for read mapping. GeneAbacus (all code available at *Vejnar, 2023c*) was used to create genomic profiles for creating tracks. Fragment coverage on each nucleotide was normalized to the total fragments in each sample per million fragments. For genome-wide analysis, only uniquely mapped reads (with alignment quality ≥30) were used.

## Peak calling for ChIP-seq

Peaks were called using MACS2 *Zhang et al., 2008* for ChIP-seq data. Narrow peaks were called using MACS2 with the additional parameters '-f BEDPE `--nomodel` --keep-dup all' with the default significance cut-off (q=0.05, high threshold) and p=0.05 (low threshold). Peaks that are called at high threshold in one condition but not called at low threshold in the other condition are defined to be specific to the condition. Genes with promoter regions (+/-1 kb of transcription start site) that overlap with a peak are defined to be associated with that peak. Tracks for PolII ChIP-seq at 5dpf of wild type or *linc-mipep; linc-wrb* mutant brains are publicly available at https://www.giraldezlab.org/data/tornini_et_al_2023_elife/.

## Single nuclei preparation for scMultiome

Flash-frozen pooled brains were prepared based on Protocol CG000366 – Rev D (Protocol 2) from 10 x Genomics (available here). It is critical to keep samples cold and/or on ice for all steps. Briefly, all samples were processed identically and simultaneously to minimize batch effects. Chilled 0.1 X

Lysis Buffer (500 µl) was immediately added to frozen samples, and samples were homogenized using a glass dounce tissue grinder with glass pestle. Samples were incubated on ice for 5 min, gently pipetted 10 x, then incubated again for 5 min. Chilled Wash Buffer (500 µl) was gently added to samples. After pipetting the mix 5 x, the samples were passed through 70µm-porosity Flowmi tips into new ice-cold low-bind 1.5 ml tubes. Each suspension was subsequently passed through a 40µm-porosity Flowmi tip into a new ice-cold low-bind 1.5 ml tube. Samples were centrifuged at 500 rcf 5 min at 4 °C. The supernatant was gently removed without disturbing the nuclei pellet. Chilled Wash Buffer (1 ml) was added, and the nuclei were gently resuspended 5 x. This wash and resuspension step was repeated one more time. On the final step, nuclei were resuspended in Diluted Nuclei Buffer. Quality and number of nuclei (as assessed by >90% Trypan Blue staining and almost no cell clumps) for each sample was assessed using a hemocytometer and were immediately used for tagmentation step using the 10 x Genomics platform. Library preparation was performed following the standard 10 x Genomics protocol (available here).

## Data analysis of scRNA-seq and scATAC-seq

Single nuclei from brains of wild type or *linc-mipep* mutant siblings were collected as described above (*Brain collection for molecular analyses*). The raw 10 x Genomics Multiome data of scRNA-seq and scATAC-seq were processed using the 10 x Genomics cellranger-arc pipeline (v1.0.1) with the genome, GRCz11. The total numbers of sequenced read pairs per sample for RNA and ATAC were between 197,900,000 and 268,400,000. The estimated numbers of cells for WT and mutant were 7,137 and 7,872, respectively. The mean numbers of raw read pairs per cell were (1) 27,742.56 for RNA and 37,593.97 for ATAC in WT and (2) 26,154.78 for RNA and 27,382.86 for mutant. The median numbers of genes per cell for WT and mutant were 349 and 365, respectively. ATAC median high-quality fragments per cell for WT and mutant were 10,466 and 8,626, respectively.

For downstream analyses, we used the Weighted Nearest Neighbor (WNN) method in Seurat (*Hao et al., 2021*). The two experimental conditions of WT and mutant were first analyzed separately. Data filtering was based on visual inspection of data distributions. The number of RNA read counts per cell was filtered between 50 and 3000 for WT and between 50 and 5000 for mutant. The number of ATAC read counts per cell was filtered between 500 and 50,000 for WT and between 500 and 80,000 for mutant. The filtering threshold for mitochondrial fractions was 15% for both WT and mutant data. Other parameters were left to default values in Seurat (v4.0.2). The numbers of filtered cells in WT and mutant were 6942 and 7740, respectively. The numbers of filtered ATAC peaks in WT and mutant were 164,266 and 167,925, respectively. We then followed the standard Seurat pipelines, with default parameters, for RNA analysis (SCTransform and PCA) and ATAC analysis (TFIDF and SVD) to obtain a WNN graph as a weighted combination of RNA and ATAC data for each of WT and mutant data. Dimensionality reduction was done by UMAP, clustering by the shared nearest neighbor and smart local moving algorithms, and differential marker identification by Wilcoxon rank sum tests. For analyses of variation in chromatin accessibility and enriched motifs, we used chromVAR (*Schep et al., 2017*) and all motifs from the *Fornes et al., 2020* database. We also performed a merged analysis of the two conditions in a similar way by merging the two datasets using the *merge* function in Seurat. We did not make any correction for batch effects because the two conditions did not show any distinct batch effects on UMAP plots of the merged data. Cell states, or types, were identified by cross-referencing with known markers on ZFIN and 5 dpf datasets from *Raj et al., 2020*.

For identification of condition-specific significant ATAC peaks in each cluster, intensity distributions of each peak in WT and mutant were statistically analyzed by the Wilcoxon rank sum and the Kolmogorov-Smirnov (KS) methods using one-tailed tests for each condition. Based on manual inspection of p-value distributions of all peaks, we chose raw p-value thresholds of 0.001 and 0.01 for the Wilcoxon and the KS tests, respectively, to deem peaks to be significant. No p-value correction was performed at this filtering step as a strategy of choice. Those significant peaks were further analyzed to identify enriched motifs as described above. In addition, for those clusters of interest, Clusters 8, 35, 38, 39, and 42, we performed a simulation for the number of significant peaks in each cluster by generating 1000 random peak intensity datasets by shuffling the intensity values between WT and mutant as many as the number of cells in the cluster in question. This simulation provided empirical null distributions of the number of significant peaks to obtain p-values. R code for data processing and analyses is available on GitHub (*Lee, 2023*).

The cells included after filtering from the Seurat analysis were used to perform integrated diffusion and MELD to keep the analyzed dataset consistent. These new techniques were implemented to analyze the data from a different approach. Integrated diffusion was used to combine multimodal datasets, specifically each cell's RNA-seq and ATAC-seq data, to create a joint data diffusion operator. The 3D integrated PHATE was computed on this joint data diffusion operator as described previously (*Kuchroo et al., 2022*; *Kuchroo et al., 2021*). To color the plots by likelihood of a cell belonging to the wildtype or mutant sample, this integrated diffusion operator was used for MELD, outputting the likelihood score for each cell belonging to a wildtype or mutant sample. The notebook for this analysis is available on GitHub (*Du, 2023*).

## Acknowledgements

We thank Dr. Shawna Hiley and Dr. Ilil Carmi for editorial and scientific input; Dr. Kaya Bilguvar, Christopher Castaldi, and Dr. Guilin Wang from the Yale Center for Genome Analysis for sequencing support; Dr. Kaelyn Sumigray for sharing Leica confocal; Dr. Mayssa Mokalled for sharing animal transgenic lines; Dr. Marcus Ghosh for code used in the F0 behavioural data analysis and for teaching F.K. the approach; and Dr. Sumru Bayin, Dr. Sarah Ackerman, and members of the Giraldez and Rihel labs for critical feedback. Research reported in this publication was supported by a K99/R00 Pathway to Independence Award from the US NIH Eunice Kennedy Shriver Institute for Child Health and Human Development (K99HD105001) and a fellowship from the Hartwell Foundation (VAT), Wellcome Trust Investigator Award 217150/Z/19/Z (JR), and Simons Foundation grant and NIH grants R01 HD100035 and MH118554 (AJG). The content is solely the responsibility of the authors and does not necessarily represent the official views of the National Institutes of Health or any funding sources. We acknowledge the Zebrafish Information Network (ZFIN).

## Additional information

### Competing interests

Smita Krishnaswamy: Reviewing editor, *eLife*. The other authors declare that no competing interests exist.

### Funding

| Funder | Grant reference number | Author |
|---|---|---|
| Eunice Kennedy Shriver National Institute of Child Health and Human Development | K99HD105001 | Valerie A Tornini |
| Hartwell Foundation | Postdoctoral fellowship | Valerie A Tornini |
| Wellcome Trust | 217150/Z/19/Z | Jason Rihel |
| Simons Foundation Autism Research Initiative | | Antonio J Giraldez |
| National Institute of Mental Health | MH118554 | Antonio J Giraldez |
| Eunice Kennedy Shriver National Institute of Child Health and Human Development | HD100035 | Antonio J Giraldez |

The funders had no role in study design, data collection and interpretation, or the decision to submit the work for publication. For the purpose of Open Access, the authors have applied a CC BY public copyright license to any Author Accepted Manuscript version arising from this submission.

## Author contributions
Valerie A Tornini, Conceptualization, Data curation, Formal analysis, Supervision, Funding acquisition, Validation, Investigation, Visualization, Methodology, Writing – original draft, Writing – review and editing; Liyun Miao, Data curation, Formal analysis, Investigation, Methodology; Ho-Joon Lee, Software, Formal analysis, Investigation, Visualization, Methodology; Timothy Gerson, Valeria Schmidt, Data curation, Investigation; Sarah E Dube, Data curation, Methodology; François Kroll, Software, Formal analysis, Visualization, Methodology; Yin Tang, Formal analysis, Investigation, Methodology; Katherine Du, Manik Kuchroo, Formal analysis, Investigation, Visualization, Methodology; Charles E Vejnar, Resources, Software, Visualization; Ariel Alejandro Bazzini, Data curation, Investigation, Visualization; Smita Krishnaswamy, Resources, Supervision, Methodology; Jason Rihel, Resources, Software, Supervision, Visualization, Methodology, Writing – review and editing; Antonio J Giraldez, Conceptualization, Resources, Supervision, Funding acquisition, Project administration, Writing – review and editing

## Author ORCIDs
Valerie A Tornini  http://orcid.org/0000-0003-2877-6057
Ho-Joon Lee  http://orcid.org/0000-0003-3616-5387
François Kroll  http://orcid.org/0000-0001-9908-2648
Charles E Vejnar  http://orcid.org/0000-0002-7132-4534
Ariel Alejandro Bazzini  http://orcid.org/0000-0002-2251-5174
Jason Rihel  http://orcid.org/0000-0003-4067-2066
Antonio J Giraldez  http://orcid.org/0000-0002-6823-137X

## Ethics
Fish lines were maintained in accordance with the AAALAC research guidelines, under a protocol approved by the Yale University Institutional Animal Care and Use Committee (IACUC Protocol Number 2021-11109). We have complied with all relevant ethical regulations under this protocol.

## Decision letter and Author response
Decision letter https://doi.org/10.7554/eLife.82249.sa1
Author response https://doi.org/10.7554/eLife.82249.sa2

# Additional files

## Supplementary files
• Supplementary file 1. Information on sORFs identified within lincRNAs and targeting/genotyping information (3 sheets).

• Supplementary file 2. Protein and proximal 3'UTR BLAST results, and related HMGN1 across species (5 sheets).

• Supplementary file 3. Correlating Drugs to *linc-mipep1* heterozygous and homozygous mutants, from hierarchical clustering analysis against>500 FDA-approved small molecules (from *Rihel et al., 2010*), and concentrations used (1 sheet).

• Supplementary file 4. Bulk omni-ATAC-seq on WT or *linc-mipep;linc-wrb* mutant brains at 5 dpf (4 sheets).

• Supplementary file 5. Single cell Multiome Analyses of WT or linc-mipep mutant brains (sibling-matched) at 5 dpf (8 sheets).

• Supplementary file 6. Integrated Diffusion/MELD analyses using WNN clusters and conditional clusters (2 sheets).

• Supplementary file 7. RNA Polymerase II ChIP-seq on wild type (WT) or *linc-mipep; linc-wrb* dissected brains at 5days post-fertilization (dpf) (2 sheets).

• Supplementary file 8. ATAC peak intensity plots for statistically different peaks between wild type and *linc-mipep* mutant cells.

• Supplementary file 9. Key for raw sequencing data from this study deposited in NCBI BioProject PRJNA945049 (1 sheet).

• MDAR checklist

## Data availability

The sequencing datasets generated and analyzed in this study have been made available through the Gene Expression Omnibus (GEO) database (Project ID PRJNA945049). The plasmids, custom antibodies, and fish lines generated in this study are available from the corresponding authors on request. Plasmids will be deposited through Addgene (202543: ubb:linc-mipep and 202544: ubb:hHmgn1). Fish lines have been requested for submission to ZIRC for distribution. Sequences used to generate ribosome profiling plots were previously published (*Bazzini et al., 2014*; *Johnstone et al., 2016*) and are available through Sequence Read Archive (SRA) with accession numbers SRP034750 and at SRP072296. All code generated and used in this study is available through GitHub repositories. Links with code are provided in each respective methods section, and as follows: Multi-frame Ribo-seq and mRNA-seq visualization (*Vejnar, 2023a*); Micropeptides_fingerprints (*Kroll, 2022*); Sleep tracking analysis (*Rihel, 2023*); LabxPipe (*Vejnar, 2023b*); GeneAbacus (*Vejnar, 2023c*); Single cell multiome analyses (*Lee, 2023*); Zebrafish Integrated Analysis (*Du, 2023*).

The following dataset was generated:

| Author(s) | Year | Dataset title | Dataset URL | Database and Identifier |
|---|---|---|---|---|
| Tornini VA, Miao L, Lee H-J, Gerson T, Dube SE, Schmidt V, Kroll F, Tang Y, Du K, Kuchroo M, Vejnar CE, Bazzini AA, Krishnaswamy S, Rihel J, Giraldez AJ | 2023 | linc-mipep and linc-wrb encode micropeptides that regulate chromatin accessibility in vertebrate-specific neural cells | https://www.ncbi.nlm.nih.gov/bioproject/PRJNA945049 | NCBI BioProject, PRJNA945049 |

The following previously published datasets were used:

| Author(s) | Year | Dataset title | Dataset URL | Database and Identifier |
|---|---|---|---|---|
| Bazzini AA, Johnstone TG, Christiano R, Mackowiak SD, Obermayer B, Fleming ES, Vejnar CE, Lee MT, Rajewsky N, Walther TC, Giraldez AJ | 2014 | Identification of small ORFs in vertebrates using ribosome footprinting and evolutionary conservation | https://www.ncbi.nlm.nih.gov/geo/query/acc.cgi?acc=GSE53693 | NCBI Gene Expression Omnibus, GSE53693 |
| Johnstone TG, Bazzini AA, Giraldez AJ | 2015 | Upstream ORFs are prevalent translational repressors in vertebrates | https://www.ncbi.nlm.nih.gov/sra/?term=SRA314809 | NCBI Sequence Read Archive, SRA314809 |
| Giraldez Lab | 2014 | Identification of small ORFs in vertebrates using ribosome footprinting and evolutionary conservation | https://www.ncbi.nlm.nih.gov/sra/?term=SRP034750 | NCBI Sequence Read Archive, SRP034750 |
| Bazzini AA, Del Viso F, Moreno-Mateos MA, Johnstone TG, Vejnar CE, Qin Y, Yao J, Khokha MK, Giraldez AJ | 2016 | Codon optimality and mRNA decay in zebrafish and Xenopus | https://www.ncbi.nlm.nih.gov/sra/?term=SRP072296 | NCBI Sequence Read Archive, SRP072296 |

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

# Appendix 1

## Appendix 1—key resources table

| Reagent type (species) or resource | Designation | Source or reference | Identifiers | Additional information |
|---|---|---|---|---|
| Gene (*Danio rerio*) | si:ch73-1a9.3, linc-mipep (also called lnc-rps25) - now hmgn1b | Ensembl | ENSDARG00000103919 | |
| Gene (*Danio rerio*) | si:ch73-281n10.2, linc-wrb - now hmgn1a | Ensembl | ENSDARG00000097102 | |
| Gene (*Homo sapiens*) | Hmgn1 | Ensembl | ENSG00000205581 | |
| Genetic reagent (*Danio rerio*) | linc-mipep[del1.78kb] | This paper | Mutant line | ya126, available from Giraldez Lab; submitted through ZIRC |
| Genetic reagent (*Danio rerio*) | linc-mipep[ATG-del6] | This paper | Mutant line | ya127, available from Giraldez Lab; submitted through ZIRC |
| Genetic reagent (*Danio rerio*) | linc-mipep[del8] | This paper | Mutant line | ya128, available from Giraldez Lab; submitted through ZIRC |
| Genetic reagent (*Danio rerio*) | linc-mipep[3'UTR-del74] | This paper | Mutant line | ya129, available from Giraldez Lab; submitted through ZIRC |
| Genetic reagent (*Danio rerio*) | linc-wrb[del11] | This paper | Mutant line | ya130, available from Giraldez Lab; submitted through ZIRC |
| Genetic reagent (*Danio rerio*) | Tg(olig2:egfp)[vu12] | *Shin et al., 2003* | transgenic line | Previously published line |
| Genetic reagent (*Danio rerio*) | Tg(ubb:linc-mipep-FLAG-HA-T2A-mCherry) | This paper | Transgenic line | ya145, available from Giraldez lab; submitted through ZIRC |
| Genetic reagent (*Danio rerio*) | Tg(ubb:human-Hmgn1-FLAG-HA-T2A-mCherry) | This paper | Transgenic line | ya151, available from Giraldez Lab; submitted through ZIRC |
| Antibody | rabbit polyclonal anti-Linc-wrb | This paper | Custom antibody | custom antibody, (1:100–200) for antibody staining; works with ProK or acetone permeabilization. |
| Antibody | Rabbit polyclonal anti-Linc-mipep | This paper | Custom antibody | custom antibody, (1:100–200)for antibody staining; works with ProK or acetone permeabilization. |
| Antibody | mouse monoclonal anti-FLAG | Sigma | Cat #:F3165 | Western blot (1:2000) |
| Antibody | rabbit polyclonal Actin | Sigma | Cat #: A5060 | Western blot (1:2000) |
| Antibody | rabbit polyclonal anti-RNA Polymerase II antibody | Abcam | Cat #: ab817 | ChIP-seq (4µg) |
| Recombinant DNA reagent | ubb:linc-mipep-FLAG-HA-T2A-mCherry | This paper | Plasmid | Available from Giraldez Lab |
| Recombinant DNA reagent | ubb:humanHmgn1-FLAG-HA-T2A-mCherry | This paper | Plasmid | Available from Giraldez Lab |
| Peptide, recombinant protein | EnGen Spy Cas9 NLS (Cas9 protein) | New England Biolabs | Cat #: M0646T | |
| Sequence-based reagent | gBlocks | Integrated DNA Technologies (IDT) | Gene blocks | Sequences in materials section |
| Sequence-based reagent | All synthetic guide RNAs | Synthego | | See *Supplementary file 1* |
| Sequence-based reagent | primers for genotyping and qPCR probes | Sigma | | see *Supplementary file 1*, and materials section |
| Sequence-based reagent | primers for RNA in situ hybridization probes | Sigma | | see *Supplementary file 1* |
| Commercial assay or kit | Neurobasal Medium | Thermo Fisher Scientific | Cat #: 21103049 | |
| Commercial assay or kit | B-27 Supplement (50X), serum free | Thermo Fisher Scientific | Cat #: 17504044 | |

*Appendix 1 Continued on next page*

*Appendix 1 Continued*

| Reagent type (species) or resource | Designation | Source or reference | Identifiers | Additional information |
|---|---|---|---|---|
| Commercial assay or kit | Monarch RNA Cleanup Kit | New England Biolabs | Cat #: T2040L | |
| Commercial assay or kit | DIG RNA Labeling Mix | Roche | Cat #: 11277073910 | |
| Commercial assay or kit | NBT/BCIP Stock Solution | Roche | Cat #: 11681451001 | |
| Commercial assay or kit | EZ-Tn5 Transposase | Lucigen | Cat #: TNP92110 | |
| Commercial assay or kit | Anti-Digoxigenin-AP, Fab fragments | Roche | Cat #: 11093274910 | |
| Commercial assay or kit | NEBNext High-Fidelity 2X PCR Master Mix | New England Biolabs | Cat #: M0541 | |
| Commercial assay or kit | Agencourt AMPureXP beads | Beckman Coulter Genomics | Cat #: A63881 | |
| Commercial assay or kit | Flowmi Cell Strainers, porosity 70µm | Bel-Art SP Scienceware | Cat #: H13680-0070 | |
| Commercial assay or kit | Flowmi Cell Strainers, porosity 40µm | Bel-Art SP Scienceware | Cat #: H13680-0040 | |
| Commercial assay or kit | Trizol Reagent | Trizol Reagent | Cat #: 15596–018 | |
| Commercial assay or kit | Nuclei Buffer* (20X) | 10x Genomics | Cat #: 2000153/2000207 | |
| Commercial assay or kit | Nonidet P40 (NP40) Substitute | Sigma-Aldrich | Cat #: 74385 | |
| Commercial assay or kit | NuPAGE 4 to 12%, Bis-Tris, 1.0–1.5mm, Mini Protein Gels | Thermo Fisher Scientific | Cat #: NP0322BOX | |
| Commercial assay or kit | NuPAGE MOPS SDS Running Buffer | Thermo Fisher Scientific | Cat #: NP0001 | |
| Commercial assay or kit | 10X Phosphate-Buffered Saline (PBS), pH 7.4 | American Bio | Cat #: AB11072-01000 | |
| Commercial assay or kit | Amplitaq DNA Polymerase | Applied Biosystems | Cat #: N8080153 | |
| Commercial assay or kit | SuperScript III Reverse Transcriptase | Invitrogen | Cat #: 18080044 | |
| Commercial assay or kit | SuperScript III Reverse Transcriptase | Invitrogen | Cat #: 18080044 | |
| Commercial assay or kit | MinElute Kit | Qiagen | Cat #: 28004 | |
| Commercial assay or kit | Chromium Single Cell Multiome ATAC + Gene Expression | 10x Genomics | 10x Genomics | |
| Chemical compound, drug | Trizma Hydrochloride Solution, pH 7.4 | Sigma-Aldrich | Cat #: T2194 | |
| Chemical compound, drug | Sodium Chloride Solution, 5M | Sigma-Aldrich | Cat #: 59,222C | |
| Chemical compound, drug | Magnesium Chloride Solution, 1M | Sigma-Aldrich | Cat #: M1028 | |
| Chemical compound, drug | L-701,324 | Tocris Bioscience | Cat #: 0907 | dissolved in DMSO |
| Chemical compound, drug | Flumethasone | Selleck Chem | Cat #: S4088 | dissolved in DMSO |
| Chemical compound, drug | Tricaine-S Topical Anesthetics | Pentair Aquatic Eco-Systems | Cat #: TRS1 | |

*Appendix 1 Continued on next page*

*Appendix 1 Continued*

| Reagent type (species) or resource | Designation | Source or reference | Identifiers | Additional information |
|---|---|---|---|---|
| Chemical compound, drug | Triton X –100 | Sigma-Aldrich | Cat #: T9284 | |
| Chemical compound, drug | Tween-20 | Sigma-Aldrich | Cat #: P1379 | |
| Chemical compound, drug | Digitonin (5%) | Thermo Fisher Scientific | Cat #: BN2006 | |
| Chemical compound, drug | DAPI | Thermo Fisher Scientific | Cat #: D1306 | |
| Chemical compound, drug | 16% Paraformaldehyde aqueous solution | Electron Microscopy Sciences | Electron Microscopy Sciences | |
| Chemical compound, drug | cOmplete, EDTA-free Protease Inhibitor Cocktail | Roche | | |
| Chemical compound, drug | T7 RNA Polymerase | Roche | Cat #: RPOLT7-RO | |
| Chemical compound, drug | Glycoblue | Thermo Fisher Scientific | Cat #: AM9516 | |
| Software, algorithm | ZebraLab | ViewPoint Behavior Technology | | http://viewpoint.fr/en/p/software/zebralab-zebrafish-behavior-screening |
| Software, algorithm | MATLAB toolboxes | MathWorks | | |
| Software, algorithm | MATLAB R2018a | MathWorks | | http://mathworks.com/products/matlab.html |
| Software, algorithm | Prism 9 | GraphPad | | https://www.graphstats.net/graphpad-prism |
| Software, algorithm | LabxDB seq | *Vejnar and Giraldez, 2020* | | Used for managing high-throughput sequencing data |
| Software, algorithm | LabxPipe | *Vejnar, 2023b* | | available at https://github.com/vejnar/LabxPipe |
| Software, algorithm | ReadKnead | *Vejnar, 2023c* | | available at https://github.com/vejnar/ReadKnead |
| Software, algorithm | Bowtie2 | *Langmead and Salzberg, 2012* | | read mapping |
| Software, algorithm | BEDTools | *Quinlan and Hall, 2010* | | genome tracks |
| Software, algorithm | MACS3 and MACS2 | *Zhang et al., 2008* | | peak calling |
| Software, algorithm | DESeq2 | *Love et al., 2014* | | differential analysis |
| Software, algorithm | deeptools | *Ramírez et al., 2014* | | |
| Software, algorithm | gplots | Galili 2020 | | available at https://github.com/talgalili/gplots |
| Software, algorithm | MEME suite | *McLeay and Bailey, 2010* | | available at https://meme-suite.org/meme/tools/ame |
| Software, algorithm | GeneAbacus | *Vejnar, 2023c* | | available at https://github.com/vejnar/geneabacus |
| Software, algorithm | cellranger-arc pipeline (v1.0.1) | 10x Genomics | | |
| Software, algorithm | Weighted Nearest Neighbor (WNN) | *Hao et al., 2021* | | |
| Software, algorithm | Integrated Diffusion | *Kuchroo et al., 2021; Kuchroo et al., 2022* | | |
| Software, algorithm | Custom sleep analysis software | *Rihel, 2023* | | available at https://github.com/JRihel/Sleep-Analysis/tree/Sleep-Analysis-Code |

