## [Editor Report]

The study describes the discovery of two related micro-peptides that regulate zebrafish behavior by affecting chromatin accessibility in the embryonic brain. Zebrafish mutants lacking these micro-peptides show altered gene regulatory networks that preferentially affect oligodendrocytes and cerebellar cells in the embryonic brain. The data presented in the study is solid and presents convincing additional evidence for versatile functions of micro-peptides.

---

## [Decision Letter]

**Decision letter after peer review:**

Thank you for submitting your article "*linc-mipep* and *linc-wrb* encode micropeptides that regulate chromatin accessibility in vertebrate-specific neural cells" for consideration by *eLife*. Your article has been reviewed by 4 peer reviewers, one of whom is a member of our Board of Reviewing Editors, and the evaluation has been overseen by and Marianne Bronner as the Senior Editor. The reviewers have opted to remain anonymous.

Essential revisions:

1) The evolutionary analysis should be expanded significantly which will increase the scope of the results. What happens in other fish species (teleosts but also coelacanth/gar)? Do they also have both proteins? What happens in frogs/birds/reptiles? A multiple-alignment showing the proteins from different representative species of HMGN1 and the new proteins will be particularly informative.

2) In the initial screen, it is not clear how the candidates for testing were selected and what kind of mutations were introduced in the F0, and what was the efficiency of the editing. As the paper is presented at least in part as an innovative screening effort, it is important to provide these details and outline them in the Results section.

3) A ChIP-seq experiment of the new proteins appears to be very interesting, but it is basically not described at all. How many peaks were found? Do they resemble each other? How reproducible was the data? A motif-based analysis appears to be very superficial given how instrumental these data (if solid) can be.

4) The authors should show ribosome profiling data together with the gene structure of examined transcript (ideally, supported by RNA-seq) to visualize the position of ribosome-protected regions within the transcripts (Extended data Figure 1a and Figure 1d). The sequence analyses reveal the similarity between linc-mipep and linc-wrb and should be presented as it is an important finding. The authors should indicate the (expected/predicted) size of both peptides; it was not mentioned in the manuscript.

5) The different genetic alleles generated for linc-mipep and linc-wrb should be confirmed by DNA sequencing chromatographs; the expression of the linc-mipep and linc-wrb transcripts in the mutants should be confirmed by qRT-PCR as sometimes even small deletions can lead to destabilization or overexpression of the remaining transcripts. This is particularly important for the mutants that show behavioral deviations from wt animals.

6) In an elegant rescue experiment, the authors demonstrate that CDS of linc-miprep can rescue zebrafish locomotion hyperactivity phenotype. A control experiment with a construct expressing a frameshifted peptide should be included. From the presentation in Figure 2a, the peptide was tagged with FLAG-HA. Can the expression of the peptide be detected by Western blot/immunostaining? Have the authors tried to rescue the phenotype with human HMGN1?

7) One of the main conclusions from this study is that both micropeptides act together/somewhat redundantly, which would explain why knocking out both peptides has a stronger phenotype than knocking out either peptide individually. While this is a possibility (that they act redundantly, targeting the same regions in the genome), other scenarios are possible, e.g. that they have distinct or only partially overlapping chromatin targets and thus regulate different genes/pathways, which in the end converge on the same behavioral phenotype.

To resolve this, the rescue with linc-mipep should be attempted for the double mutant and also the single linc-wrb mutant (since it is a ubiquitous overexpression line, it may rescue both). Similarly, a rescue by linc-wrb (which is not shown, also not for the single mutant) would be important to support the conclusion that the phenotype is due to the loss of this peptide, and that it acts redundantly with linc-mipep. Moreover, it will also be important to quantify and provide statistics for the overexpression effect of the rescue construct in the WT background

Please also address the other points raised by the reviewers to improve the clarity and readability of the manuscript.

*Reviewer #1 (Recommendations for the authors):*

In my opinion, the main weakness of the paper is the very limited ability of the molecular phenotypic characterization of the mutants to explain the behavioral and neuropharmacological phenotype. This weakness is partially evident also by the lack of this point in the discussion that focuses on the evolutionary implications and the chromatin remodeling defects observed in the mutants. This is in my opinion an important point that should be better explained and investigated.

I would have also liked to have some validation of the protein localization in the cell types identified as most sensitive to the loss of linc-mipep and linc-wrb. Custom antibodies for these peptides were generated and staining is presented in extended fig2m showing only the larval forebrain. This analysis should be extended to OPC and cerebellar granule cells.

In the discussion of the putative evolutionary origin of linc-mipep and linc-wrb the authors mention the lancelet defining it simply as "invertebrate". This polyphyletic group is insufficient here and the authors should explain better its relevance in this context as basal chordate.

*Reviewer #2 (Recommendations for the authors):*

1. In the initial screen, it is not clear how the candidates for testing were selected and what kind of mutations were introduced in the F0, and what was the efficiency of the editing. As the paper is presented at least in part as an innovative screening effort, it is important to provide these details and outline them in the Results section.

2. The evolutionary analysis can be expanded significantly which will increase the scope of the results. What happens in other fish species (teleosts but also coelacanth/gar)? Do they also have both proteins? What happens in frogs/birds/reptiles? A multiple-alignment showing the proteins from different representative species of HMGN1 and the new proteins will be particularly informative.

3. Locomotor activity graphs: the number of tested fish should be added to all graphs. In some cases, the authors added a dot plot graph with P values, and this should be done for all the locomotor activity experiments.

4. The rescue experiments were performed using zebrafish linc-mipep CDS. It would be interesting to test whether a homolog for a different species (i.e., HMGN1) will also rescue the behavioral phenotypes.

5. ATAC-seq analysis: the analysis focuses on the comparison of peaks detected or not detected in the different datasets. A more common and more robust approach is to identify a single set of peaks using all the data together, and then test (e.g., using DESeq2) which peaks have differential accessibility between the different genotypes/samples.

6. A ChIP-seq experiment of the new proteins appears to be very interesting, but it is basically not described at all. How many peaks were found? Do they resemble each other? How reproducible was the data? A motif-based analysis appears to be very superficial given how instrumental these data (if solid) can be.

7. There's a mistake in c-fos In situ hybridization experiment location, which is in extended data Figure 4E, and not in Figure 3f (where it is written now).

8. In figure 2d – is the phenotype of linc-mipep-/- vs. linc-mipep+/+ fish (1st vs. 3rd) here significant? If yes – show the p-value. If not – how is this explained?

9. The statement that genes with ribosome-protected fragments are likely encoding functional proteins is not always correct and this part should be explained in more detail.

10. In the description of the single-cell datasets, please indicate fold-changes in differences of representation (e.g., for reduction of olig2+ oligodendrocyte progenitor cells across the brain)

*Reviewer #3 (Recommendations for the authors):*

1. The authors should show ribosome profiling data together with the gene structure of examined transcript (ideally, supported by RNA-seq) to visualize the position of ribosome-protected regions within the transcripts (Extended data Figure 1a and Figure 1d). The sequence analyses reveal the similarity between linc-mipep and linc-wrb and should be presented as it is an important finding. The authors should indicate the (expected/predicted) size of both peptides; it was not mentioned in the manuscript.

2. The authors should elaborate on the expression of the examined transcripts/peptides during embryogenesis (i.e., are they expressed at 5dpf only or earlier/later) and in adult tissues.

3. The different genetic alleles generated for linc-mipep and linc-wrb should be confirmed by DNA sequencing chromatographs; the expression of the linc-mipep and linc-wrb transcripts in the mutants should be confirmed by qRT-PCR as sometimes even small deletions can lead to destabilization or overexpression of the remaining transcripts. This is particularly important for the mutants that show behavioral deviations from wt animals.

4. In an elegant rescue experiment, the authors demonstrate that CDS of linc-miprep can rescue zebrafish locomotion hyperactivity phenotype. A control experiment with a construct expressing a frameshifted peptide should be included. From the presentation in Figure 2a, the peptide was tagged with FLAG-HA. Can the expression of the peptide be detected by Western blot/immunostaining? Have the authors tried to rescue the phenotype with human HMGN1?

5. A question related to the comment above: is it possible to detect native, untagged peptides by mass spectrometry? Have the authors tried to do it?

6. The manuscript would gain on clarity if a more detailed description of the behavioral assays used as a functional read-out was included in the main text. In general, the manuscript is partially hard to follow due to the insufficient data presentation, peptide size, peptide sequences, etc.

7. The authors should elaborate on why they used a single linc-mipep mutant for the drug experiments but a double mutant for omni-ATAC experiments.

8. The authors should clearly state in the discussion that the molecular mechanisms of action of both studied peptides remain completely unknown. For example, how do they affect chromatin accessibility? What are their interaction partners if any? etc

*Reviewer #4 (Recommendations for the authors):*

The manuscript can be significantly improved by addressing the following concerns:

Concerns and suggestions:

One of the main conclusions from this study is that both micropeptides act together/somewhat redundantly, which would explain why knocking out both peptides has a stronger phenotype than knocking out either peptide individually. While this is a possibility (that they act redundantly, targeting the same regions in the genome), other scenarios are possible, e.g. that they have distinct or only partially overlapping chromatin targets and thus regulate different genes/pathways, which in the end converge on the same behavioral phenotype.

To reconcile this, the rescue with linc-mipep should be attempted for the double mutant and also the single linc-wrb mutant (since it is a ubiquitous overexpression line, it may rescue both). Similarly, a rescue by linc-wrb (which is not shown, also not for the single mutant) would be important to support the conclusion that the phenotype is due to loss of this peptide, and that it acts redundantly with linc-mipep. Moreover, it will also be important to quantify and provide statistics for the overexpression effect of the rescue construct in the WT background – is there a significant activity decrease by linc-mipep OE? Overall, the authors mention the dosage-sensitivity of HMGN1 proteins, but with the current analyses fail to provide convincing evidence of a clear dosage effect of the two peptides since they could potentially target different, only in part redundant, genes or have different effects in different cell types. To this end, the use of either the single linc-mipep vs double linc-mipep/linc-wrb mutant is inconsistent in the second half of the manuscript: global ATAC-Seq data is only provided from the double mutant while single-cell-analyses are only provided from the single linc-mipep mutant. Moreover, the ChIP-seq analyses provided are only summarized for both proteins combined in the main Figure, but used individual antibodies, leaving it unclear how the individual profiles look (the authors should follow the standard convention on how to show the quality of ChIP-seq data, e.g. provide ChIP-seq tracks at least for some example genes since the quality of the data remains unclear, and differences between the two Abs cannot be assessed; the Suppl Table 8 also only provides a combined list of 37 genes for which ChIP seq peaks were identified though it would be important to show it individually for each AB; also the number of genes bound appears really really small? Are these ALL genes with a ChIP-seq peak?).

The second major concern relates to the unclear link between the different phenotypes observed: how can the behavioral phenotypes be reconciled with the molecular phenotypes (chromatin accessibility in specific neurons or precursors), and how can the chromatin accessibility differences in WT vs mutant be reconciled with the measured transcriptional/gene expression differences? Is there any evidence for NMDA being downstream of linc-mipep/wrb regulation? I applaud the authors on generating all these interesting data sets and analyses, but without connecting them together (here the focus for example on just the single linc-mipep mutant would be helpful, but the global brain ATAC-Seq data is only shown for the double mutant; and vice-versa, the single-cell ATAC-Seq data with the chromatin accessibility changes detected in specific cell types is not linked back to the ChIP-seq profiles of the peptides). Do glial progenitors and OPCs of the mutant(s) have altered expression of the underlying loci with altered accessibility? In Figure 4c, e, f, h, and Extended 6c-e, how does the chromatin accessibility translate to rna level in Purkinje cells and radial glia cells? How many sites lose accessibility in OPCs? Is "broad loss" a fair assessment of the observation?

Without addressing the two major concern points, the statement that linc-mipep and linc-wrb 'broadly regulate the chromatin state of neural cell types, most impacting OPCs and cerebellar granule cell gene expression networks and cell states in a basal vertebrate' appears overstated and would need to phrased differently/softened.

---

## [Author Response]

Essential revisions:(1) The evolutionary analysis should be expanded significantly which will increase the scope of the results. What happens in other fish species (teleosts but also coelacanth/gar)? Do they also have both proteins? What happens in frogs/birds/reptiles? A multiple-alignment showing the proteins from different representative species of HMGN1 and the new proteins will be particularly informative.

We have performed in depth evolutionary analysis. We observe that other teleosts species have two copies of these genes, while coelacanth and gar have only one copy. Frogs/birds/reptiles also have only one copy of these genes.

Detailed Response: We performed a Clustal Omega multiple-alignment showing the full-length alignment of proteins for key species (including coelacanth, gar, *Xenopus,* zebra finch, and anole lizard). We have also included these data for all analyzed species in Supplementary Table 2 (tab 5). This analysis provides information on key conserved sequences across species. Additionally, we have provided data on syntenic relationships across species and have included a syntenic alignment diagram to replace the previous version that only included human and zebrafish orthologues. We also clarify in the text, within figure supplements, and supplementary tables which of these are currently known, annotated, or otherwise annotated as noncoding or pseudogenes.

Location of new data: A protein sequence alignment among select vertebrate species (including coelacanth and gar) is presented in Figure 2 —figure supplement 4. The synteny analysis (including spotted gar) is presented in Figure 2 —figure supplement 5. Information on these genes in other fish species (including teleosts) is included in Figure 2 —figure supplement 6.

Additional information from all identified related sequences is presented in Supplementary Table 2 (especially sheet 5).

(2) In the initial screen, it is not clear how the candidates for testing were selected and what kind of mutations were introduced in the F0, and what was the efficiency of the editing. As the paper is presented at least in part as an innovative screening effort, it is important to provide these details and outline them in the Results section.

Candidates were selected for testing by analyzing ribosome profiles of highly expressed transcripts at 12hpf, 24 hpf, or 48 hpf, targeted these genes to test for CRISPR lethality, and in situ hybridization was performed on 21 of the remaining candidates. From these genes, we identified brain-enriched micropeptides on which we focused for further screening.

F0 mutations varied, with some generating small in- and out-of-frame indels, and some generating larger mutations. All guides efficiently edited their target sequences, with indel or large deletion rates with multiple guides estimated between ~40-100%. The efficiency of the editing ranged, depending on guide efficiencies.

Detailed Response: We have included details about how candidates for testing were selected in the main text (page 2) and in Supplementary Table 1 (tab 2). Briefly, we analyzed ribosome profiles for previously published lincRNAs in zebrafish embryos during early development (0-24 hours post-fertilization) (Bazzini et al., 2014), performed preliminary CRISPR/Cas9 targeting and excluded from further study those that were lethal. We then performed in situ hybridization on 21 of these candidates and identified brain-enriched micropeptide candidates, which we focus on for further screening.

To determine the efficiency of editing and what kinds of mutations were introduced in F0s, we pooled 8 representative F0 larvae for each of the tested guides and performed Inference of CRISPR Edits (ICE, Synthego) analysis to identify how efficiently the guides cut and their estimated indel or large deletion rates. All genes were edited, with some target guide combinations inducing small indels, and some inducing also larger deletions. We have updated the methods section to reflect these additions, specifically. We further discuss some limitations for targeting small transcripts, which may have limited GC-rich sequences that traditional CRISPR/Cas9 target sites require, and PAM-less variants available now.

Location of new data: Data showing efficient CRISPR targeting and some example edited sequences is presented in Figure 1 —figure supplement 2, with descriptions in Results section on page 2. Information on in situ hybridization screen results are presented in Supplementary Table 1 (sheet 2) and described in Results section on page 2.

(3) A ChIP-seq experiment of the new proteins appears to be very interesting, but it is basically not described at all. How many peaks were found? Do they resemble each other? How reproducible was the data? A motif-based analysis appears to be very superficial given how instrumental these data (if solid) can be.

Thank you for the comments. We found 315 peaks, yet we could not analyze if they resemble each other because our ChIP-seq combined the antibodies for both proteins to identify common bound regions using the double maternal-zygotic mutants as controls. To address the reviewer’s comments, we have performed replicate ChIP-seq for each protein in two different experiments to characterize the antibodies independently for ChIP-seq of Linc-wrb and Linc-mipep; however, given the low number of peaks identified (357 for Linc-wrb and 78 for Linc mipep) we are concerned about the validity of these peaks and the application of those antibodies for ChIP-seq. Given these results, we believe that future studies will be needed to perform an in-depth characterization of the binding profile of each individual protein, the regions bound in the chromatin, and the developmental progression of their binding profile in the brain and in other tissues, thus we believe this is beyond the current scope of the paper and have removed this analysis.

Detailed Response: In our previous submission, we provided a combined ChIP-seq analysis of both proteins at 24 hpf using both antibodies pooled together to identify overlapping binding regions. As a control we had used a double-maternal-zygotic mutant embryos at 24 hpf (over each respective input). The results we presented in the previous submission, through which we performed the motif-based analysis, represented 315 peaks that were called in wild type compared to double MZ *linc-mipep; linc-wrb* mutant embryos. To address the reviewer’s comments, we have performed replicates in two different experiments to characterize the antibodies independently for ChIP-seq. First, we performed a ChIP-seq for each individual antibody, using 4 hpf wild type embryos (for which our lab has optimized the ChIP-seq protocol) and double-mutant 4 hpf embryos as controls (over each respective input). At this 4 hpf timepoint, we found an enrichment of nuclear antibody staining in non-dividing cells, as shown in Figure 2 —figure supplement 2I and J. While we were able to call statistically significant peaks (357 peaks for Linc-wrb (q=0.05) and 78 peaks for Linc-mipep (q=0.05)), visual inspection of the tracks representing the ChIP-seq profile did not provide a clear ChIP-seq signal enrichment at all called peaks that would make us confident of the validity of these results. Please see Author response image 1. Second, because we had been able to detect the protein by probing with an anti-FLAG antibody in *ubb:linc-mipep-FLAG-HA-T2A-mCherry* embryos (which has a FLAG tag on the C-terminal end of the *linc-mipep* CDS), we reasoned that we may be able to perform ChIP-seq analyses using this transgenic *ubb:linc-mipep-FLAG-HA-T2A-mCherry* which expresses linc-mipep-FLAG protein as assessed by western blot. We performed ChIP-seq analyses using a FLAG antibody for 4 hpf embryos from *ubb:linc-mipep-FLAG-HA-T2A-mCherry* incrosses (to ensure maternally-deposited transcripts and enrichment for protein at early stages) and wild type embryos as a control (over each respective input). These analyses did not result in obvious peaks upon visual inspection of called peaks. Based on these attempts, and our inability to definitively identify peaks that are clearly enriched compared to the control, we have decided to remove the previously submitted results from our analyses and interpretations, and from this report. We believe these results, which were performed at 4 hpf and 24 hpf, do not affect the overall conclusions of the paper which are centered at 5-6 dpf, because we clearly observed differentially regulated regions in chromatin accessibility and gene expression between the wild type and mutant brains at 5-6 dpf. Future efforts will be needed to further explore the chromatin binding profile of these proteins in the brain and other tissues.

**Author response image 1. sa2fig1:** 

(4) The authors should show ribosome profiling data together with the gene structure of examined transcript (ideally, supported by RNA-seq) to visualize the position of ribosome-protected regions within the transcripts (Extended data Figure 1a and Figure 1d). The sequence analyses reveal the similarity between linc-mipep and linc-wrb and should be presented as it is an important finding. The authors should indicate the (expected/predicted) size of both peptides; it was not mentioned in the manuscript.

We now include ribosome footprint and ribosome-depleted RNA-seq tracks (at 48 hpf) for the *linc-mipep* and *linc-wrb* genes, in Figure 1 —figure supplement 5D and E. We present the sequences in Figure 2E and have expanded the evolutionary analysis across vertebrate species, in Figure 2 —figure supplements 3 and 5, and Supplementary Table 2. For clarity, we also added the sizes of proteins encoded by *linc-mipep (87 aa)* and *linc-wrb (93 aa)* to Figure 2E and have added it to the main text (pages 3 and 4).

(5) The different genetic alleles generated for linc-mipep and linc-wrb should be confirmed by DNA sequencing chromatographs; the expression of the linc-mipep and linc-wrb transcripts in the mutants should be confirmed by qRT-PCR as sometimes even small deletions can lead to destabilization or overexpression of the remaining transcripts. This is particularly important for the mutants that show behavioral deviations from wt animals.

We now include DNA sequencing chromatographs for each of the alleles, now presented in Figure 1 —figure supplement 6, alongside their respective behavioral profiles. These data confirm the generation of *linc-mipep* and *linc-wrb* mutations. Antibody staining revealed a loss of protein staining (presented in Figure 2 —figure supplement 2) and analysis of the RNA levels in scRNA seq in *linc-mipep* mutant brain cells (presented in Figure 4 —figure supplement 2) reveal that the transcript is strongly reduced likely due to nonsense-mediated decay.

(6) In an elegant rescue experiment, the authors demonstrate that CDS of linc-miprep can rescue zebrafish locomotion hyperactivity phenotype. A control experiment with a construct expressing a frameshifted peptide should be included. From the presentation in Figure 2a, the peptide was tagged with FLAG-HA. Can the expression of the peptide be detected by Western blot/immunostaining? Have the authors tried to rescue the phenotype with human HMGN1?

We demonstrate that human *HMGN1* can rescue linc-miprep mutants *(*Figure 2F and G)*.* To test whether the CDS of a related human protein, *HMGN1,* would be sufficient to rescue the phenotype of *linc-mipep* and *linc-wrb,* we generated a stable transgenic *ubb:human-Hmgn1-FLAG-HA-T2A-mCherry (ubb:hHmgn1)* line. We found that when we crossed this *ubb:hHmgn1* line to *linc-mipep* mutants, we were able to significantly rescue the *linc-mipep* hyperactivity. These results are presented in Figure 2F and G.

We can detect expression of the peptide FLAG- *linc-mipep* expressing line by western blot, confirming a protein-coding rescue. This data is presented in Figure 2 —figure supplement 1A.

We considered a frame-shifted CDS overexpression experiments, but felt that the results may be difficult to interpret, for example if overexpression of a frameshifted peptide resulted in novel behavioral phenotypes. We note that, although the specific overexpression constructs are different than used here, Chiu et al. (2016) performed a large sleep/wake behavioral screen on the effects of over-expressing 1286 ORFs, of which most gave no phenotype. They further tested 60 overexpression lines in stable transgenic lines, and found only 12 had behavioral phenotypes, spread across sleep-wake parameters, with some increasing, some decreasing, and some having no effect on activity. Based on this data, over-expressing constructs in general are not expected to have consistent non-specific effects on locomotor activity. Furthermore, because we only use the CDS of *linc-mipep or HMGN1* in the transgenic rescue experiments (excluding 5’ and 3’UTR sequences), and the mutants for *linc*-*mipep1* and *linc-wrb* include an 8nt and 1 nt frameshift deletions, respectively, plus the long generation time to achieve the above mentioned experiment with the frameshift rescue, we hope that the reviewers find the data presented here sufficient evidence to support the function of these coding genes.

(7) One of the main conclusions from this study is that both micropeptides act together/somewhat redundantly, which would explain why knocking out both peptides has a stronger phenotype than knocking out either peptide individually. While this is a possibility (that they act redundantly, targeting the same regions in the genome), other scenarios are possible, e.g. that they have distinct or only partially overlapping chromatin targets and thus regulate different genes/pathways, which in the end converge on the same behavioral phenotype.To resolve this, the rescue with linc-mipep should be attempted for the double mutant and also the single linc-wrb mutant (since it is a ubiquitous overexpression line, it may rescue both). Similarly, a rescue by linc-wrb (which is not shown, also not for the single mutant) would be important to support the conclusion that the phenotype is due to the loss of this peptide, and that it acts redundantly with linc-mipep. Moreover, it will also be important to quantify and provide statistics for the overexpression effect of the rescue construct in the WT background.

We thank the reviewer for these suggestions. As we mentioned above (comment 6), we were able to rescue *linc-mipep* mutants with a human *Hmgn1* transgene (Figure 2F and G). However, this transgene did not rescue *linc-wrb* (Figure 2 —figure supplement 1D). Yet, we found that *ubb:linc-mipep* rescues *linc-wrb* heterozygous mutants almost to wild type levels (p=0.058) (Figure 2 —figure supplement 1B and C), supporting at least a partially redundant function of these proteins.

While we would like to attempt a rescue of the double mutant (*linc-mipep; linc-wrb*) with the *linc-mipep* CDS*,* this would require at least 2 generations equivalent to at least an additional 6 months, and we do not think the number of animals required for this experiment, based on a 3Rs ethical perspective, justifies this experiment which we believe would not significantly change the conclusions of the paper based on the new results presented in this revision.

As suggested by the reviewers, we have provided statistics for the *linc-mipep* overexpression effect of the rescue construct in the WT background (Figure 2B).

Moreover, we now include data on intermediate phenotypes in larvae resulting from *linc-mipep;linc-wrb* double-heterozygous crosses, in Figure 1 —figure supplement 7C and D. These analyses reveal that each mutation causes very similar hyperactivity levels and behavioral profiles. We think these results suggest a dose-dependent effect, reflected in the observed levels of hyperactivity in heterozygous and homozygous mutants. We have therefore included a discussion point on the potential individual, overlapping, or redundant effects of each of these genes.

Location of new data: The *linc-mipep* rescue of the *linc-wrb* mutant phenotype is presented in Figure 2 —figure supplement 1B and C. Statistics for *linc-mipep* overexpression in wild type backgrounds is included in Figure 2B. A behavior plot and dot plot from a double-heterozygous incross showing dosage-dependent phenotypes are presented in Figure 1 —figure supplement 7C and D.

Reviewer #1 (Recommendations for the authors):In my opinion, the main weakness of the paper is the very limited ability of the molecular phenotypic characterization of the mutants to explain the behavioral and neuropharmacological phenotype. This weakness is partially evident also by the lack of this point in the discussion that focuses on the evolutionary implications and the chromatin remodeling defects observed in the mutants. This is in my opinion an important point that should be better explained and investigated.

We have further analyzed and discussed the molecular phenotypic characterization of the mutants to explain the behavioral and neuropharmacological phenotypes. Using scRNA-seq and ATAC-seq data, we have now identified gene expression changes, as well as corresponding changes in chromatin accessibility, in Purkinje and OPC cells. We also show that some of these changes affect genes important for NMDA and glucocorticoid signaling activity. These results suggest that loss of *linc-mipep* leads to dysregulation of multiple genes that more strongly affect oligodendrocytes and cerebellar cell types, including genes important for the activity of NMDA and glucocorticoid signaling pathways.

Detailed response: We have further analyzed the molecular phenotypes of affected cells to understand the link between the behavioral and neuropharmacological phenotypes. These analyses suggest a role for *linc-mipep* in the regulation of genes required for OPC and cerebellar cell type development, including NMDA receptor and glucocorticoid receptor signaling components.

We first searched for changes in gene expression in the cell types of interest – OPCs, cerebellar granule cells, and Purkinje cells – that may indicate how these cells were impacted. From single-cell data, we found that *linc-mipep* mutant Purkinje cells showed significantly decreased expression of numerous genes, including *roraa, Rorb, foxp4*, and *prkcg*, which are required for maturation or maintenance of Purkinje cells in zebrafish. We also identified numerous genes that were dysregulated in wild type OPCs relative to *linc-mipep* mutants. Some of these genes, including *erbb4b, mag, qkia,* and *myt1b* were also enriched in cerebellar granule cells, pointing to similar gene networks that may be disrupted in these affected cell types.

We then assessed omni-ATAC-seq data at some of the differentially expressed genes in OPCs. We found differentially accessible regions in *linc-mipep*;*linc-wrb* mutant brains downstream of *olig2,* within a large intronic span of *sgms2b,* and upstream of *fabp7a*, suggesting that the micropeptides may be required for proper gene regulation in OPCs.

Finally, we asked whether some of the genes that are differentially expressed or regulated between mutant and wild-type OPC, cerebellar granule cell, or Purkinje cells that may explain the dysregulation of NMDA and the sensitization to glucocorticoids that we observed in the mutants (from Figure 3). Indeed, in *linc-mipep; linc-wrb* brains, we found increased accessibility at two genes associated with glucocorticoid downstream signaling (*fkbp5* in OPCs*)* and stress responses *(scg5* in granule cells*)*; the expression of these two genes is also downregulated in *linc-mipep* mutants. We also observed decreased expression in the *linc-mipep* OPC cluster of numerous genes involved with NMDA receptor activity (*aldocb, ttyj3b, slc1a2b, nrxn1a, grin1b, gpmbaa, atp1a1b).*

Finally, focusing on NMDA receptor regulation, we identified two regions exhibiting differential chromatin accessibility within *grin1b*, a gene encoding an NMDAR subunit*.* At the single-cell level, expression of *grin1b* (and, to some degree also *grin1a*, which encodes another subunit)*,* is significantly higher in wild type OPCs relative to *linc-mipep* mutant OPCs.

These new results are consistent with a model in which loss of *linc-mipep* leads to dysregulation of multiple genes in oligodendrocytes and cerebellar cell types, including genes important for the activity of NMDA and glucocorticoid signaling pathways. We have also included descriptions of these results in main text (pages 8-9) and detailed discussion about these points (page 10). Additional work, beyond the scope of this manuscript, will be needed to test the specific contribution of each of these changes to the neuronal and behavioral effects of *linc-mipep* and *linc-wrb* mutations.

Location of new data: These new data and analyses are presented in Figure 4H; Figure 4 —figure supplements 5, 6, 7, and 8; and Supplementary Table 4.

I would have also liked to have some validation of the protein localization in the cell types identified as most sensitive to the loss of linc-mipep and linc-wrb. Custom antibodies for these peptides were generated and staining is presented in extended fig2m showing only the larval forebrain. This analysis should be extended to OPC and cerebellar granule cells.

We performed antibody staining in *olig2:GFP* brains at 5 dpf to characterize the protein localization of proteins encoded by *linc-mipep* and *linc-wrb* in OPCs and cerebellar regions. We note that we do not have a good antibody or transgenic line to label cerebellar granule cells, though we present antibody staining of Linc-mipep and Linc-wrb in single Z confocal slices of brains at 5 dpf, which show a generally uniform expression pattern for both proteins. We find that the Linc-mipep antibody signal is stronger compared to Linc-wrb (also notable in Figure 2 —figure supplement 2E and G), though Linc-mipep is weakly expressed in the torus longitudinalis and tegmentum (as in Figure 4 —figure supplement 4B and E). We added images of whole brains in *olig2:GFP* and wild type fish with these antibodies. These new data are presented in Figure 4 —figure supplement 6.

In the discussion of the putative evolutionary origin of linc-mipep and linc-wrb the authors mention the lancelet defining it simply as "invertebrate". This polyphyletic group is insufficient here and the authors should explain better its relevance in this context as basal chordate.

We clarify in the text the lancelet’s relevance in this context as a basal chordate (on page 4). We have also now added context for the synteny analysis, in Figure 2 —figure supplement 5A and Supplementary Table 2. We further clarify in the Results section that these findings are consistent with what is known about HMGN family members and highlight that our analysis identifies the putative HMGN origin in lamprey, which seems to be derived partially from the N-terminal sequence of a protein-coding gene in the lancelet, presented in Figure 2 —figure supplement 3D.

Reviewer #2 (Recommendations for the authors):1. In the initial screen, it is not clear how the candidates for testing were selected and what kind of mutations were introduced in the F0, and what was the efficiency of the editing. As the paper is presented at least in part as an innovative screening effort, it is important to provide these details and outline them in the Results section.

Please see response to comment #2 above, and copied below.

2. The evolutionary analysis can be expanded significantly which will increase the scope of the results. What happens in other fish species (teleosts but also coelacanth/gar)? Do they also have both proteins? What happens in frogs/birds/reptiles? A multiple-alignment showing the proteins from different representative species of HMGN1 and the new proteins will be particularly informative.

Please see response to comment #1 above, and copied below.

3. Locomotor activity graphs: the number of tested fish should be added to all graphs. In some cases, the authors added a dot plot graph with P values, and this should be done for all the locomotor activity experiments.

We have included the number of fish in all the locomotor activity graphs, and have also added dot plot graphs with P values for all the locomotor activity experiments.

Location of new data: Number of fish for locomotor activity graphs are now included in Figure 1 G and H, Figure 1 —figure supplement 6A through E, Figure 1 —figure supplement 7A and D; Figure 2B, C, and F; Figure 2 —figure supplement 1 B and D; Figure 3B; Figure 3 —figure supplement 2A; Figure 3 —figure supplement 2A, D, and F. Dot plots have been added to all relevant locomotor activity graphs except Figure 1 —figure supplement 6, because that data is mostly represented in Figure 1 – supplementary figure 7A , C, and D.

4. The rescue experiments were performed using zebrafish linc-mipep CDS. It would be interesting to test whether a homolog for a different species (i.e., HMGN1) will also rescue the behavioral phenotypes.

We have generated a stable ubiquitous overexpression line with the human HMGN1 CDS, and have assessed behavioral phenotypes in wild type and *linc-mipep* or *linc-wrb* mutant backgrounds. We found that overexpression of the human HMGN1 CDS is sufficient to rescue the phenotype in *linc-mipep* mutants. Please also see Comment #6 and 7 above.

Location of new data: These results are presented in Figure 2F and G, and in Figure 2 —figure supplement 1B and C.

5. ATAC-seq analysis: the analysis focuses on the comparison of peaks detected or not detected in the different datasets. A more common and more robust approach is to identify a single set of peaks using all the data together, and then test (e.g., using DESeq2) which peaks have differential accessibility between the different genotypes/samples.

We provide an updated ATAC-seq analysis using 3 replicates, identifying a single set of peaks using all the data together, and then testing using DESeq which peaks have differential accessibility between wild-type and *linc-mipep;linc-wrb* mutant brains at 5 dpf. We find that the new results are consistent with the previous analyses and provide a more robust and refined interpretation to identifying differentially accessible peaks between wild type and mutant brains. These analyses more strongly support some of the findings in this study, including the transcription factor motifs identified as enriched or depleted in mutants, and specific peaks identified as differentially accessible.

Location of new data: These data are provided in Figure 3D and E; Figure 3 —figure supplement 3 A through C; sample tracks in Figure 4 —figure supplement 6A through F; and updated Supplementary Table 4. We have also updated the link to publicly available tracks for these runs.

6. A ChIP-seq experiment of the new proteins appears to be very interesting, but it is basically not described at all. How many peaks were found? Do they resemble each other? How reproducible was the data? A motif-based analysis appears to be very superficial given how instrumental these data (if solid) can be.

Please see response to Comment #3 above,

7. There's a mistake in c-fos In situ hybridization experiment location, which is in extended data Figure 4E, and not in Figure 3f (where it is written now).

Thank you for catching this error. We have corrected this mistake, and have indicated the correct experiment location, now in Figure 3 —figure supplement 3D.

8. In figure 2d – is the phenotype of linc-mipep-/- vs. linc-mipep+/+ fish (1st vs. 3rd) here significant? If yes – show the p-value. If not – how is this explained?

The phenotype of *linc-mipep-/- vs.* WT fish in Figure 2D is significant (p = 0.031, Dunnett’s test). We have now included the P values in the graph.

9. The statement that genes with ribosome-protected fragments are likely encoding functional proteins is not always correct and this part should be explained in more detail.

We have added a more detailed explanation, in the Results section (page 3) and Discussion section (page 11).

10. In the description of the single-cell datasets, please indicate fold-changes in differences of representation (e.g., for reduction of olig2+ oligodendrocyte progenitor cells across the brain).

We have added a more detailed explanation, in the Results section (page 3) and Discussion section (page 11).

Reviewer #3 (Recommendations for the authors):1. The authors should show ribosome profiling data together with the gene structure of examined transcript (ideally, supported by RNA-seq) to visualize the position of ribosome-protected regions within the transcripts (Extended data Figure 1a and Figure 1d). The sequence analyses reveal the similarity between linc-mipep and linc-wrb and should be presented as it is an important finding. The authors should indicate the (expected/predicted) size of both peptides; it was not mentioned in the manuscript.

We now include tracks with ribosome footprints and ribosome-depleted RNA-seq above the gene structure of each examined transcript, inFigure 1 —figure supplement 5D and E. We also highlight that sequence analyses reveal similarity between *linc-mipep* and *linc-wrb* as an important finding in the Results section (page 4), with extended evolutionary analyses presented in Figure 2 —figure supplements 3 – 6 and Supplementary 2. We have now added the size of both peptides to the text (pages 3 and 4) and directly in Figure 2E.

2. The authors should elaborate on the expression of the examined transcripts/peptides during embryogenesis (i.e., are they expressed at 5dpf only or earlier/later) and in adult tissues.

*linc-mipep* and *linc-wrb* transcripts are expressed starting from the 1-cell state zygote stage, throughout early development, with protein expression assessed starting at 4 hpf through 5-6 dpf.

Location of new data: We have included data on the expression of both transcripts, in Figure 1 —figure supplement 5F and G. We also include whole embryo antibody staining for each of the protein, in Figure 2 —figure supplement 2. We did not examine adult tissues, as we focused our studies on early (neuro)developmental stages.

3. The different genetic alleles generated for linc-mipep and linc-wrb should be confirmed by DNA sequencing chromatographs; the expression of the linc-mipep and linc-wrb transcripts in the mutants should be confirmed by qRT-PCR as sometimes even small deletions can lead to destabilization or overexpression of the remaining transcripts. This is particularly important for the mutants that show behavioral deviations from wt animals.

Please see response to Comment #3 above.

4. In an elegant rescue experiment, the authors demonstrate that CDS of linc-miprep can rescue zebrafish locomotion hyperactivity phenotype. A control experiment with a construct expressing a frameshifted peptide should be included. From the presentation in Figure 2a, the peptide was tagged with FLAG-HA. Can the expression of the peptide be detected by Western blot/immunostaining? Have the authors tried to rescue the phenotype with human HMGN1?

Please see response to Comment #6.

5. A question related to the comment above: is it possible to detect native, untagged peptides by mass spectrometry? Have the authors tried to do it?

We have not attempted to perform mass spectrometry, though we expect the peptides to be detectable, as we detect them by antibody staining that is absent in the mutant embryos. We include a point in the discussion (page 11) about additional approaches beyond ribosome profiling, including mass spectrometry, to identify small peptides.

6. The manuscript would gain on clarity if a more detailed description of the behavioral assays used as a functional read-out was included in the main text. In general, the manuscript is partially hard to follow due to the insufficient data presentation, peptide size, peptide sequences, etc.

We have provided more detailed descriptions throughout the main text of the manuscript, specifically about behavioral assays (page 2), and have provided additional supporting information throughout the main figures and figure supplements about the peptide sizes (87aa and 93aa) in Figure 2E, peptide sequences in Figure 2E, Figure 2 —figure supplement 4, and Supplemental Table 2, and protein expression patterns in Figure 2 —figure supplement 2 and in Figure 4 —figure supplement 4.

7. The authors should elaborate on why they used a single linc-mipep mutant for the drug experiments but a double mutant for omni-ATAC experiments.

We elaborate in the Results section that we used single *linc-mipep* mutants for drug experiments, as we had found similar drugs that correlated with *linc-mipep* and *linc-wrb* mutant fingerprints (which we now include as data). To ensure we assessed the full loss-of-function of these two related genes, we performed omni-ATAC-seq experiments in double mutants.

Detailed response: In this revised manuscript, we now present data showing that NMDA receptor antagonism is a common pathway affected in *linc-wrb* mutants (Figure 3 —figure supplement 1B and 2D-G). To circumvent batch effects from unmatched (non-sibling) samples, and because our results so far indicated generally overlapping functions for *linc-mipep* and *linc-wrb,* we chose to analyze *linc-mipep* mutant brain cells and validate findings in vivo in *linc-mipep; linc-wrb* double mutants. We describe this rationale on pages 6 and 7. We clarify in the Discussion section (pages 10-11) that further work will be needed to elucidate the overlapping and unique molecular roles of the proteins encoded by *linc-mipep* and *linc-wrb.*

Location of new data: We present the *linc-mipep* or *linc-wrb* mutants’ correlating fingerprints in Figure 3—figure supplement 1A and B, and note overlapping hits in blue text. We also present the results of *linc-wrb* mutants treated with either flumethasone or L-701-324 in Figure 3 —figure supplement 2D-G.

8. The authors should clearly state in the discussion that the molecular mechanisms of action of both studied peptides remain completely unknown. For example, how do they affect chromatin accessibility? What are their interaction partners if any? etc

We have elaborated in the discussion that the molecular mechanisms of these proteins, both direct and indirect, remain unknown (pages 10-11). We provide references on work done on the related Hmgn1 in mammals (page 1), and state that future work will be needed to fully elucidate the molecular mechanisms and binding/interaction partners for each protein in zebrafish.

Reviewer #4 (Recommendations for the authors):The manuscript can be significantly improved by addressing the following concerns:Concerns and suggestions:One of the main conclusions from this study is that both micropeptides act together/somewhat redundantly, which would explain why knocking out both peptides has a stronger phenotype than knocking out either peptide individually. While this is a possibility (that they act redundantly, targeting the same regions in the genome), other scenarios are possible, e.g. that they have distinct or only partially overlapping chromatin targets and thus regulate different genes/pathways, which in the end converge on the same behavioral phenotype.To reconcile this, the rescue with linc-mipep should be attempted for the double mutant and also the single linc-wrb mutant (since it is a ubiquitous overexpression line, it may rescue both). Similarly, a rescue by linc-wrb (which is not shown, also not for the single mutant) would be important to support the conclusion that the phenotype is due to loss of this peptide, and that it acts redundantly with linc-mipep. Moreover, it will also be important to quantify and provide statistics for the overexpression effect of the rescue construct in the WT background – is there a significant activity decrease by linc-mipep OE? Overall, the authors mention the dosage-sensitivity of HMGN1 proteins, but with the current analyses fail to provide convincing evidence of a clear dosage effect of the two peptides since they could potentially target different, only in part redundant, genes or have different effects in different cell types. To this end, the use of either the single linc-mipep vs double linc-mipep/linc-wrb mutant is inconsistent in the second half of the manuscript: global ATAC-Seq data is only provided from the double mutant while single-cell-analyses are only provided from the single linc-mipep mutant. Moreover, the ChIP-seq analyses provided are only summarized for both proteins combined in the main Figure, but used individual antibodies, leaving it unclear how the individual profiles look (the authors should follow the standard convention on how to show the quality of ChIP-seq data, e.g. provide ChIP-seq tracks at least for some example genes since the quality of the data remains unclear, and differences between the two Abs cannot be assessed; the Suppl Table 8 also only provides a combined list of 37 genes for which ChIP seq peaks were identified though it would be important to show it individually for each AB; also the number of genes bound appears really really small? Are these ALL genes with a ChIP-seq peak?).

We have addressed parts of this comment in Comments #3, 6, and 7 above. Please see below for copied responses per comment section:

Comment 1a: One of the main conclusions from this study is that both micropeptides act together/somewhat redundantly, which would explain why knocking out both peptides has a stronger phenotype than knocking out either peptide individually. While this is a possibility (that they act redundantly, targeting the same regions in the genome), other scenarios are possible, e.g. that they have distinct or only partially overlapping chromatin targets and thus regulate different genes/pathways, which in the end converge on the same behavioral phenotype.To reconcile this, the rescue with linc-mipep should be attempted for the double mutant and also the single linc-wrb mutant (since it is a ubiquitous overexpression line, it may rescue both). Similarly, a rescue by linc-wrb (which is not shown, also not for the single mutant) would be important to support the conclusion that the phenotype is due to loss of this peptide, and that it acts redundantly with linc-mipep. Moreover, it will also be important to quantify and provide statistics for the overexpression effect of the rescue construct in the WT background – is there a significant activity decrease by linc-mipep OE?

See Essential revisions comment 7.

Comment 1b: Overall, the authors mention the dosage-sensitivity of HMGN1 proteins, but with the current analyses fail to provide convincing evidence of a clear dosage effect of the two peptides since they could potentially target different, only in part redundant, genes or have different effects in different cell types. To this end, the use of either the single linc-mipep vs double linc-mipep/linc-wrb mutant is inconsistent in the second half of the manuscript: global ATAC-Seq data is only provided from the double mutant while single-cell-analyses are only provided from the single linc-mipep mutant.

To circumvent batch effects from unmatched (non-sibling) samples, and because our results so far indicated generally overlapping functions for *linc-mipep* and *linc-wrb,* we chose to analyze *linc-mipep* mutant brain cells and validate findings in vivo in *linc-mipep; linc-wrb* double mutants. We describe this rationale on pages 6 and 7. We clarify in the Discussion section (pages 10-11) that further work will be needed to elucidate the overlapping and unique molecular roles of the proteins encoded by *linc-mipep* and *linc-wrb.*

Comment 1c: Moreover, the ChIP-seq analyses provided are only summarized for both proteins combined in the main Figure, but used individual antibodies, leaving it unclear how the individual profiles look (the authors should follow the standard convention on how to show the quality of ChIP-seq data, e.g. provide ChIP-seq tracks at least for some example genes since the quality of the data remains unclear, and differences between the two Abs cannot be assessed; the Suppl Table 8 also only provides a combined list of 37 genes for which ChIP seq peaks were identified though it would be important to show it individually for each AB; also the number of genes bound appears really really small? Are these ALL genes with a ChIP-seq peak?).

See Essential revisions comment 3.

The second major concern relates to the unclear link between the different phenotypes observed: how can the behavioral phenotypes be reconciled with the molecular phenotypes (chromatin accessibility in specific neurons or precursors), and how can the chromatin accessibility differences in WT vs mutant be reconciled with the measured transcriptional/gene expression differences? Is there any evidence for NMDA being downstream of linc-mipep/wrb regulation? I applaud the authors on generating all these interesting data sets and analyses, but without connecting them together (here the focus for example on just the single linc-mipep mutant would be helpful, but the global brain ATAC-Seq data is only shown for the double mutant; and vice-versa, the single-cell ATAC-Seq data with the chromatin accessibility changes detected in specific cell types is not linked back to the ChIP-seq profiles of the peptides). Do glial progenitors and OPCs of the mutant(s) have altered expression of the underlying loci with altered accessibility? In Figure 4c, e, f, h, and Extended 6c-e, how does the chromatin accessibility translate to rna level in Purkinje cells and radial glia cells? How many sites lose accessibility in OPCs? Is "broad loss" a fair assessment of the observation?Without addressing the two major concern points, the statement that linc-mipep and linc-wrb 'broadly regulate the chromatin state of neural cell types, most impacting OPCs and cerebellar granule cell gene expression networks and cell states in a basal vertebrate' appears overstated and would need to phrased differently/softened.

In this revision, we have now more fully analyzed and characterized the molecular phenotypes to link them with the behavioral phenotypes. In these analyses, we more deeply connect the single-cell analyses with bulk chromatin accessibility phenotypes. To address most of this comment, we refer to our response to a very similar point made by another reviewer, which we believe address the points made in this comment. We note that for single cell experiments with sparse data, the link between accessible regions as potential enhancers and the genes affected by those enhancers is a large challenge in the field, yet in our analyses we have shown how some of the cell type-specific transcriptomic changes show neighboring chromatin accessibility changes. We also note that we have modified our language to include a “loss” of accessibility (in the most statistically significantly affected peaks) in OPCs instead of “broad loss,” to more accurately present these results.

See also Reviewer 1 comment 1.

Without addressing the two major concern points, the statement that linc-mipep and linc-wrb 'broadly regulate the chromatin state of neural cell types, most impacting OPCs and cerebellar granule cell gene expression networks and cell states in a basal vertebrate' appears overstated and would need to phrased differently/softened.

We have adjusted the language as suggested.